# A non-canonical striatopallidal Go pathway that supports motor control

Marie A. Labouesse [1,2,3,4] ✉, Arturo Torres-Herraez[1,2,15],
Muhammad O. Chohan [1,5,15], Joseph M. Villarin[1,2,15], Julia Greenwald [1,2],
Xiaoxiao Sun[2,6], Mysarah Zahran[2,7], Alice Tang[2,8], Sherry Lam[9],
Jeremy Veenstra-VanderWeele[1,5], Clay O. Lacefield [1,2], Jordi Bonaventura [9,10],
Michael Michaelides [9,11], C. Savio Chan [12], Ofer Yizhar [13] &
Christoph Kellendonk [1,2,14] ✉

In the classical model of the basal ganglia, direct pathway striatal projection neurons (dSPNs) send projections to the substantia nigra (SNr) and entopeduncular nucleus to regulate motor function. Recent studies have re-established that dSPNs also possess axon collaterals within the globus pallidus (GPe) (bridging collaterals), yet the significance of these collaterals for behavior is unknown. Here we use in vivo optical and chemogenetic tools combined with deep learning approaches in mice to dissect the roles of dSPN GPe collaterals in motor function. We find that dSPNs projecting to the SNr send synchronous motor-related information to the GPe via axon collaterals. Inhibition of native activity in dSPN GPe terminals impairs motor activity and function via regulation of Npas1 neurons. We propose a model by which dSPN GPe axon collaterals (striatopallidal Go pathway) act in concert with the canonical terminals in the SNr to support motor control by inhibiting Npas1 neurons.

In the classical model of the basal ganglia (BG) two segregated and functionally opposing pathways connect its input, the striatum, with its midbrain outputs, the substantia nigra reticulata (SNr) and entopeduncular nucleus (EP). GABAergic striatal projection neurons in the direct pathway (dSPNs) project monosynaptically to the SNr and EP and promote disinhibition of thalamo-cortical activity and locomotion (functionally known as the Go pathway). Conversely, striatal projection neurons in the indirect pathway (iSPNs) inhibit thalamo-cortical activity and locomotion (NoGo pathway) via an additional synapse in the GPe, which in turn relays to the SNr[1]. However, many recent studies have challenged both the functional dichotomy[2–5] and anatomical organization[6–9] of the classical BG model. Tracing studies have also shown that all major BG nuclei send a subset of arborized axons collateralizing within one to four target regions. These axon collaterals

[1]Department of Psychiatry, Vagelos College of Physicians and Surgeons, Columbia University Irving Medical Center, New York, NY 10032, USA. [2]Division of Molecular Therapeutics, New York State Psychiatric Institute, New York, NY 10032, USA. [3]Department of Health, Sciences and Technology, ETH Zurich, 8092 Zurich, Switzerland. [4]Neuroscience Center Zurich, ETH Zurich and University of Zurich, 8057 Zurich, Switzerland. [5]Division of Child and Adolescent Psychiatry, New York State Psychiatric Institute, New York, NY 10032, USA. [6]Department of Biomedical Engineering, Columbia University, New York, NY 10027, USA. [7]Barnard College, Columbia University, New York, NY 10027, USA. [8]Columbia College, Columbia University, New York, NY 10027, USA. [9]Biobehavioral Imaging and Molecular Neuropsychopharmacology Unit, National Institute on Drug Abuse Intramural Research Program, Baltimore, MD 21224, USA. [10]Departament de Patologia i Terapèutica Experimental, Institut de Neurociències, L'Hospitalet de Llobregat, Universitat de Barcelona, Barcelona, Spain. [11]Department of Psychiatry & Behavioral Sciences, Johns Hopkins University School of Medicine, Baltimore, MD 21205, USA. [12]Department of Neuroscience, Feinberg School of Medicine, Northwestern University, Chicago, IL 60611, USA. [13]Departments of Brain Sciences and Molecular Neuroscience, Weizmann Institute of Science, Rehovot 76100, Israel. [14]Department of Molecular Pharmacology & Therapeutics, Vagelos College of Physicians and Surgeons, Columbia University Irving Medical Center, New York, NY 10032, USA. [15]These authors contributed equally: Arturo Torres-Herraez, Muhammad O. Chohan, Joseph M. Villarin. ✉e-mail: marie.labouesse@hest.ethz.ch; ck491@cumc.columbia.edu

could help shape BG information flow in space or time by sending copies of the same signals to distinct regions[8,10,11]. However, the relative dynamics and specific roles of axon collaterals in the BG have largely been understudied, because they often represent a smaller proportion of the output and because they are technically difficult to target.

Axon collaterals arising from striatal dSPNs, also known as bridging collaterals, are a prominent example. dSPNs send terminal projections to the midbrain, but also arborize via axon collaterals (bridging collaterals) into the GPe, the classical projection area of iSPNs. Single neuron labeling studies performed in over 100 projection neurons in the rat have shown that 37% projected exclusively to the GPe (pure indirect pathway), whereas only 3% projected solely to the SNr or EP (pure direct pathway). Sixty percent of labeled neurons projected to the SNr/EP and possessed collateral terminal fields in the GPe[12–14]. Moreover, in vivo optogenetic stimulation of dSPNs was shown to inhibit activity in the GPe to half the magnitude produced by iSPN stimulation[6]. Importantly, bridging collaterals primarily target Npas1+ or FoxP2+ neurons in the GPe rather than Nkx2.1+ or parvalbumin (PV)+ neurons, which are mostly targeted by iSPNs[15–19]. This makes for an intriguing anatomical circuit because Npas1+ and/or FoxP2+ neurons do not follow the classical BG organization, as they heavily project back to the striatum[20–23]. Despite this singular anatomical organization and connectivity, the behavioral significance of bridging collaterals is still unknown.

Previously, we had observed that the density of bridging collaterals in the GPe is highly plastic in the adult animal, being regulated by dopamine D2Rs and neural excitability. High levels of bridging collaterals also lead to a stronger reduction in GPe firing rate following dSPN optogenetic stimulation[6]. Recent work extends our findings, showing that neural activity and 6-OHDA dopamine lesions modulate bridging collateral density or connectivity to the GPe[7,16,24]. These data argue for a role of bridging collaterals in shaping the output of the BG circuitry and motor function.

To more directly understand the significance of bridging collaterals for behavior, it is essential to record their activity dynamics in awake-behaving mice as well as to inhibit their activity acutely during natural behavior. Here, we overcame existing technical challenges to address these questions. We combined terminal-specific in vivo calcium photometry or manipulation techniques, with in vivo physiology, closed loop approaches, and deep learning-based behavioral tracking to dissect the role and relative dynamics of dSPN GPe bridging collaterals in motor function (summarized in Supplementary Fig. S1). Specifically, we wanted to test two alternative hypotheses: (1) bridging collaterals functionally diverge from SNr terminals, acting like a second NoGo pathway to inhibit the GPe and locomotion; or (2) they act in convergence with canonical SNr projections working as a second Go pathway to promote locomotion and support motor function.

## Results

### dSPNs terminals in the GPe (bridging collaterals) represent more than half the density of SNr terminals

To quantify the proportion of dSPN terminals in the GPe vs. SNr, Drd1-cre mice received an adeno-associated virus (AAV) expressing a flexed GCaMP6s tagged to Synaptophysin to enrich fluorescent tracer expression in presynaptic terminals[25]. AAV was injected into the dorsomedial (DMS) part of the dorsal striatum (dStr, Fig. 1A, Supplementary Fig. S2). We found that GCaMP6s expression was largely restricted to dSPN VGAT+ puncta and not fibers of passage (Fig. 1B, C) and that dSPN terminals in the GPe accounted for more than half the density of SNr terminals (Fig. 1D). We also used dual retrograde tracing (validated in Supplementary Fig. S3), finding that GPe-projecting dSPNs (labeled with green retrograde flexed HSV) colocalize with SNr-projecting dSPNs (labeled with red retrobeads) in the DMS. Quantification showed that out of 159 YFP+ cells, 147 were also retrobeads+ (92.5%) and 12 were retrobeads- (7.5%). Out of 148 retrobeads+ cells

counted, all but 1 were YFP+ (99.3%). This indicates that the majority (99%) of SNr-projecting dSPNs have collaterals in the GPe consistent with[12–14]. In addition, out of all GPe-projecting dSPNs, a small population (7.5%) does not project to the SNr, possibly corresponding to cells described in ref. 26 (Fig. 1E). Altogether, this confirms our previous work showing that most dSPN axons on their way to the SNr arborize within the GPe via bridging collaterals[6,16].

### dSPNs send copies of motor signals to the GPe and SNr, continuously encoding body speed

The motor-promoting role of the classical SNr terminals arising from DMS dSPN cells is well established[6,16,27–30]. It is unknown, however, whether DMS dSPN GPe terminals also shape motor output. To address this question, we first determined whether GPe bridging collaterals get activated during motor tasks. We also wondered if and to what extent GPe terminals receive a copy of the same neuronal information sent by dSPNs to SNr terminals. Indeed, existing work points to a highly correlated but possible dissociation of activity between soma and presynaptic terminals in various systems[31–34] and to the existence of multiple mechanisms for local regulation of presynaptic calcium levels[35,36]. In this respect, the degree to which the activity of presynaptic terminals from two different axonal outputs correlates vs. dissociates has not been addressed. Measuring the in vivo activity of presynaptic terminals is not possible with in vivo electrophysiology. On the other hand, in vivo calcium recordings with fiber photometry at terminal sites allows to record calcium dynamics as a proxy for presynaptic activity[33,37]. We therefore used dual fiber photometry to concurrently record calcium activity in dSPN GPe and SNr axon arbors. dSPN neurons are known to track multiple motor variables across a variety of behavioral modalities[3,38]. Here we chose to record the activity of dSPN terminals in two standard behavioral assays well-known to engage and require the striatum and its direct inputs: an open field self-paced locomotion test and a rotarod motor task[33,39–46]. In a first set of experiments Drd1-cre mice were unilaterally injected with a flexed jGCaMP7s-expressing AAV[47] into the DMS and implanted with optic fibers above the GPe and SNr (Fig. 2A, B, Supplementary Figs. S2, S10). The speed of the body center speed was computed as mice locomoted in an open field arena (Fig. 2C, D, Supplementary Video S1). We used body center speed to compute the onset and offset of locomotor movements and averaged all motor bouts aligned to the onset and offset (Fig. 2D, E). This confirmed that mouse speed increased at the onset of movements and decreased at their offset. Like mouse speed, we found that dSPN activity in the GPe and SNr (quantified using the z-score of deltaF/F; dFF) increased at movement onset and decreased at movement offset (Fig. 2F). Moreover, total GPe and SNr activity correlated significantly with mouse speed (compared to shuffled control) with a maximal Pearson correlation around 0.5 (GPe: $r = 0.56$; SNr: $r = 0.51$) (Fig. 2G). These data showed that dSPN axons in the GPe and SNr continuously encode mouse speed during locomotion, consistent with findings at the cell body[43,48]. Importantly, activity of dSPN axons in the GPe and SNr were highly correlated with each other ($r = 0.78$) (Fig. 2H), suggesting they encode copies of the same neuronal information during self-paced locomotion. These data align with existing models emphasizing a role for striatal SPNs in representing the speed (or vigor) of body movements in a continuous manner[43] and indicate that such information is transmitted down to synaptic terminals. More importantly, they indicate that bridging collaterals are indeed activated during motor tasks, whereby dSPN cell bodies transfer motor information synchronously to the GPe and SNr via an axonal copy.

### dSPN GPe and SNr axons track the temporal boundaries of motor jumping bouts

We next determined whether the activity of bridging collaterals is regulated by more complex motor behaviors. Indeed, striatal SPNs are known to track multiple types of motor variables beyond body speed;

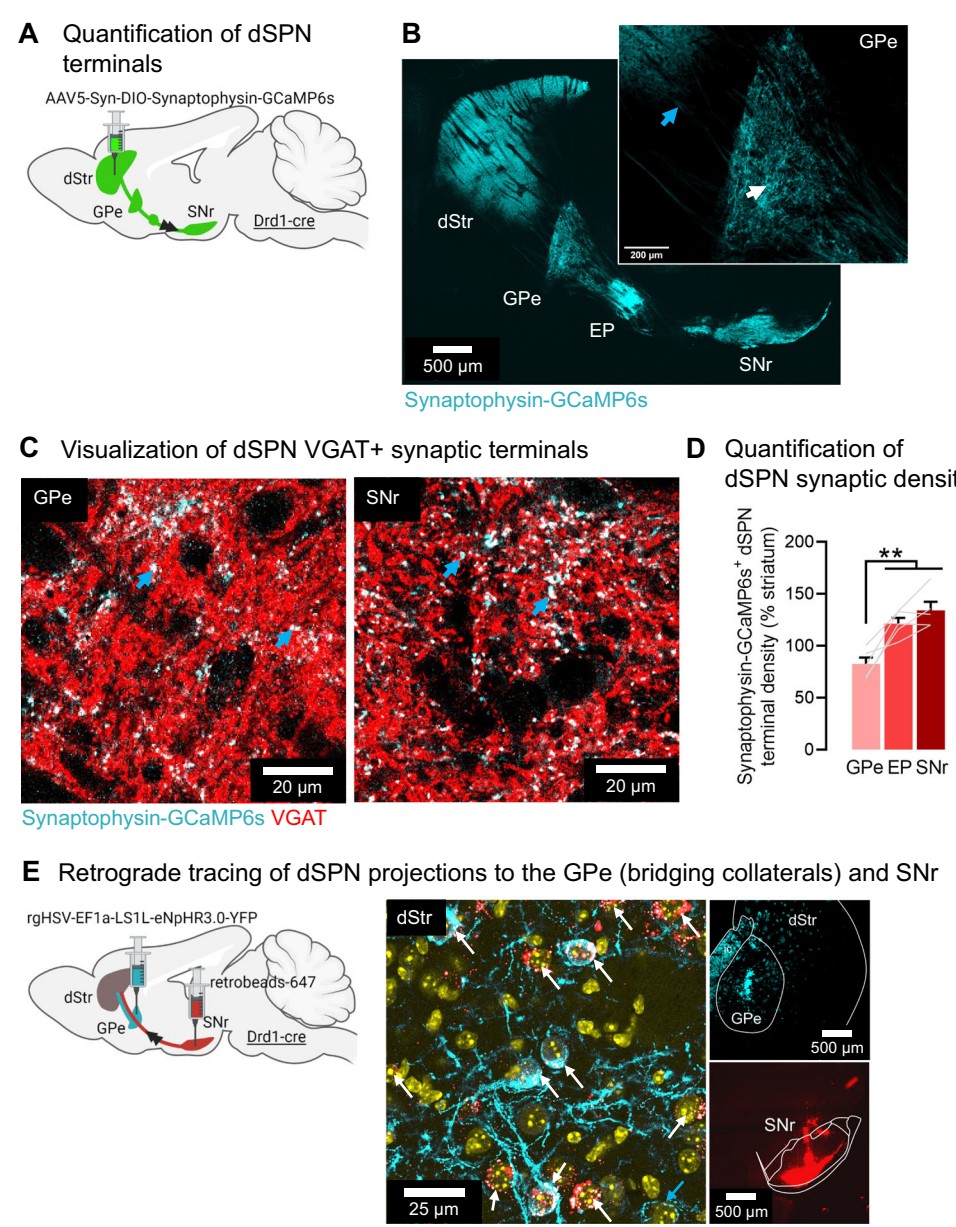

**Fig. 1 | The density of dSPN terminals in the GPe account for more than half the density of SNr terminals. A** Strategy for anterograde tracing of direct pathway striatal projection neuron (dSPN) axons/terminals using the presynapse-targeted tracer Synaptophysin-GCaMP6s. **B** Synaptophysin-GCaMP6s is largely absent from axons (blue arrow) and enriched in terminals (white arrow) (representative images from *N* = 5 mice). **C** Synaptophysin-targeted dSPN terminals (cyan) colocalize with the presynaptic GABA marker VGAT (red), appearing white (blue arrows). **D** The density of dSPN Synaptophyin-GCaMP6s+ (antibody amplified for GFP) terminals in the globus pallidus externus (GPe) (83%) reaches more than half the density in the entopeduncular nucleus (EP) (123%) and substantia nigra reticulata (SNr) (134%) (ANOVA: region *p* < 0.001; Tukey post-hocs: **\*\**p* < 0.01) (*N* = 5 mice). Data are mean ± SEM. **E** Confirmation that dSPN terminals in the GPe arise from axons

projecting to the SNr (representative images from *N* = 3 mice). Left: Injection of retrograde herpes-simplex virus (HSV) expressing a flexed YFP into the GPe and red retrobeads (retrobeads-647) into the SNr of Drd1-cre mice. Right: YFP+ cell bodies colocalized with retrobeads+ cells in the DMS, identifying dSPNs projecting to both GPe and SNr (white arrows). Note that retrobeads had a puncta-like labeling pattern, while YFP staining either had a puncta-like pattern or covered the whole soma. White or red puncta in YFP-positive soma indicate colocalization. There were also retrobeads-, YFP+ cell bodies, identifying neurons projecting only to the GPe (blue arrow). Out of 159 YFP+ cells counted, 147 were also retrobeads+ (92.5%), 12 were retrobeads- (7.5%). Out of 148 retrobeads+ cells counted, all but 1 was YFP+ (99.3%). Exact *p*-values are given in Supplementary Dataset S2. See also Supplementary Figs. S2, S3.

for instance they show sustained activity throughout the execution of motor sequences or track the temporal boundaries (onset, offset) of individual movements[40,48,49]. Mice were subjected to a rotarod motor assay known to engage the striatum[33,39–46] which allows to impose repetitive motor patterns allowing experimental control on running speed and trial averaging. First, mice were subjected to 10 rotarod trials at accelerating speed (5–40 rpm). We found that the activity of GPe and SNr axons was sustained throughout the rotarod epoch

compared to pre and post rest periods (Supplementary Fig. S4A–C), showing a higher baseline fluorescence and area under the curve (AUC) (Fig. S4D). Sustained dSPN terminal activity would be consistent with evidence for certain striatal units showing sustained activation during rotarod running[40]. When zooming in onto individual peaks, we also noticed that the properties of peaks arising from dSPN GPe or SNr axons strongly differed across task epochs, showing higher frequency and lower amplitude (taken from the local baseline) during rotarod

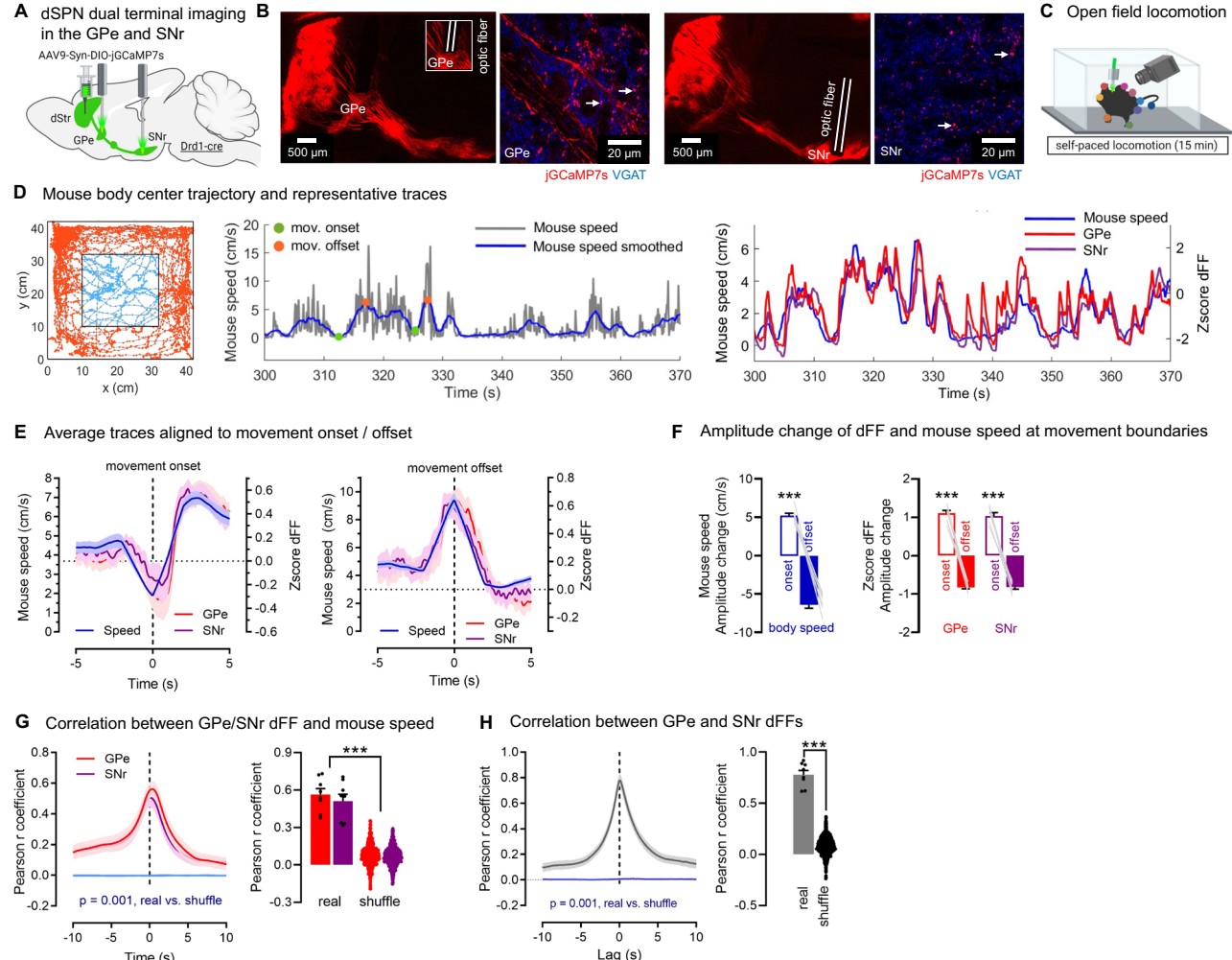

**Fig. 2 | dSPNs send copies of motor signals to GPe and SNr axons, continuously encoding body speed. A** Strategy for dual calcium recordings with fiber photometry of direct pathway striatal projection neuron (dSPN) terminals in the globus pallidus externus (GPe) and substantia nigra reticulata (SNr) arising from neurons in the dorsomedial section of the dorsal striatum (dStr) using the calcium indicator jGCaMP7s. **B** jGCaMP7s (red) colocalizes with the GABA presynaptic marker VGAT (blue) in the GPe and SNr, appearing pink (white arrows). Optic fibers target GPe and SNr jGCaMP7s+ regions. **C** Mice are video-recorded in the open field and body positions obtained with DeepLabCut. **D** Left: Representative body trajectory. Middle: Representative trace of mouse speed over time (raw data: gray; smoothed in 2 s bins: blue), showing the onset (green) and offset (orange circle) of motor bouts. Right: dSPN GPe and SNr calcium signals (Zscore of the normalized fluorescence, i.e., deltaF/F; dFF) closely track mouse speed. **E** Average data aligned to the onset and offset of individual motor bouts. GPe and SNr dFF show increases at movement onset and decreases at movement offset. **F** Body speed (two-sided paired $t$-test ***$p < 0.001$) and GPe/SNr dFF (ANOVA: main effect: ***$p < 0.001$) significantly increase at movement onset vs. offset. **G** GPe ($r = 0.56$) and SNr ($r = 0.51$) dFF significantly correlate with mouse speed when compared to phase-shuffled data ($N = 1000$ iterations) (two-sided Mann–Whitney GPe ***$p = 0.001$, SNr ***$p = 0.001$). Real data: black dots show individual mice; shuffle data: colored dots show individual shuffled data. **H** GPe and SNr dFF are highly correlated with each other (Pearson $r = 0.78$) vs. phase-shuffled data ($N = 1000$ iterations) (two-sided Mann–Whitney ***$p = 0.001$). Real data: black dots show individual mice; shuffle data: black dots show individual shuffled data. $N = 8$ mice for all panels. Exact $p$-values are given in Supplementary Dataset S2. Data are mean ± SEM. Source data are provided as a Source Data file.

running (Supplementary Fig. S4D). Since no learning-related differences emerged across the 10 trials, trials were pooled. The interval between peaks was also significantly shorter in the rotarod running epoch as opposed to pre/post rest periods (Supplementary Fig. S4E). Together, these data indicated that GPe and SNr axons are activated by running and likely track running-related motor parameters in the rotarod task.

We hypothesized that dSPN axons track the temporal boundaries (onset and offset) of task-specific body movements. To address this, we monitored the individual trajectories of mouse body parts during running using DeepLabCut[50] (Supplementary Fig. S5A). Although we did observe foot stepping behavior, it was highly variable and foot tracking quality in our setup was not good enough to allow behavior/calcium cross-analyses (Supplementary Fig S5A). However, we observed that, while performing the rotarod, all mice adopted a

behavioral strategy to jump up the rotarod then slide back down (Supplementary Video S2). This was made evident by tracking the vertical position ($Y$ axis) of the lower body, which regularly alternated between the lower and upper bounds of the rotarod (also seen in ref. 51) (Fig. 3D). To see if the duration of jumps decreased as the rotarod speed increased, we exposed animals to rotarod trials at different constant speeds (5, 10 or 15 rpm) (Fig. 3A–C). We computed the interpeak intervals of the lower body Y position between consecutive jumps, and as expected there was a significant decrease in interpeak interval with increased rotarod speeds (Fig. 3E). We then aligned the Ca2+ signal to the onset/offset of jumps to determine if GPe/SNr axons track the temporal boundaries of jumping bouts. When averaging >1000 jumping bouts per trial type (5, 10, 15 rpm) we found that dSPN GPe and SNr signals were time-locked to the boundaries of the jumps (up at onset and down at offset) (Fig. 3F, Supplementary Fig. S5B).

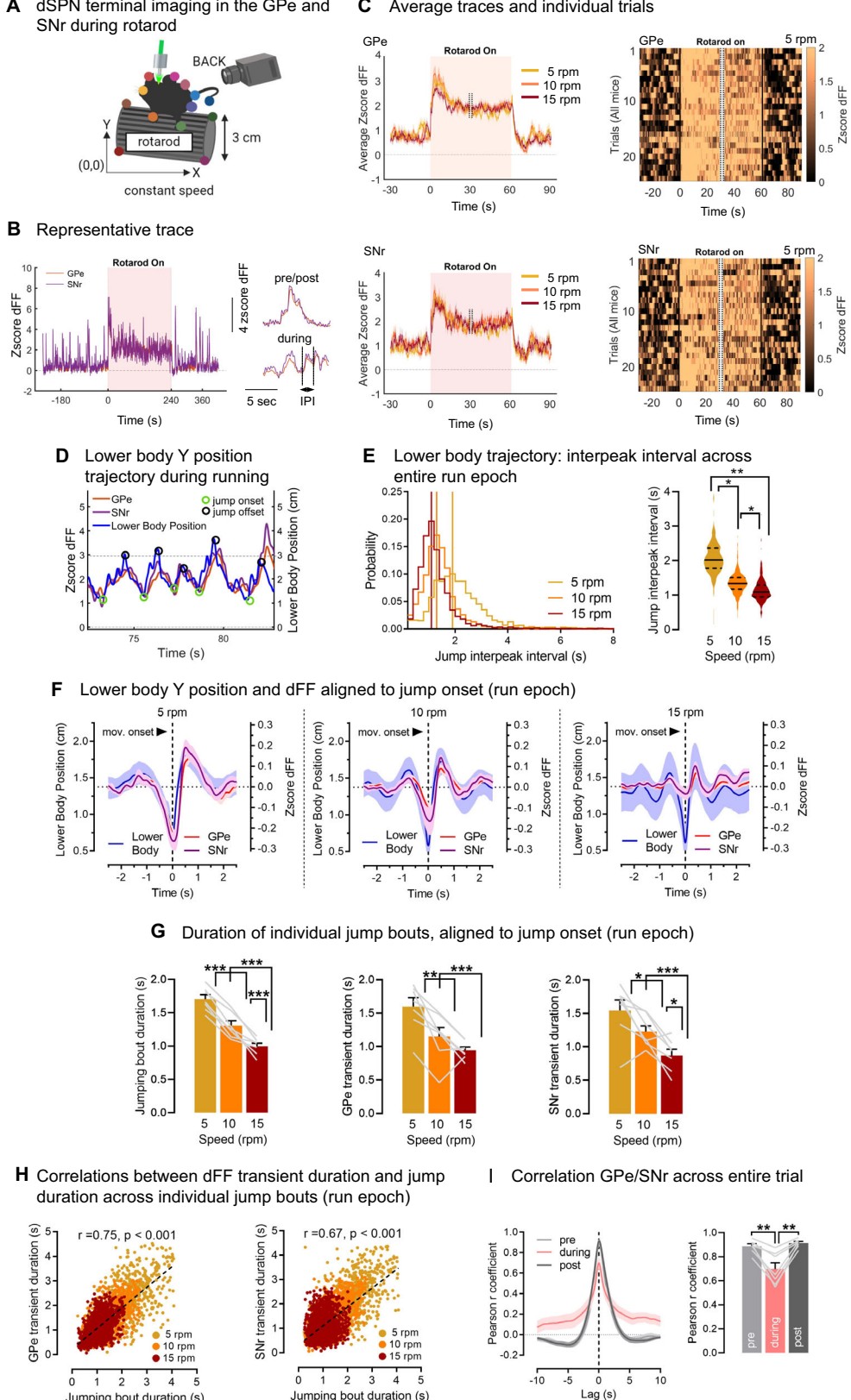

**A** dSPN terminal imaging in the GPe and SNr during rotarod

**B** Representative trace

**C** Average traces and individual trials

**D** Lower body Y position trajectory during running

**E** Lower body trajectory: interpeak interval across entire run epoch

**F** Lower body Y position and dFF aligned to jump onset (run epoch)

**G** Duration of individual jump bouts, aligned to jump onset (run epoch)

**H** Correlations between dFF transient duration and jump duration across individual jump bouts (run epoch)

**I** Correlation GPe/SNr across entire trial

Moreover, as expected, jumps became more frequent as the rotarod speed increased. Similarly, GPe/SNr jump-related transients became more frequent at increasing rotarod speeds (Fig. 3G). We then conducted statistical analyses on >4000 individual jumps, to determine if the temporal properties of individual GPe/SNr transients are adjusted on a trial-by-trial basis. We found that within individual motor bouts,

the jump bout duration (duration between two consecutive jumps) correlated significantly with the GPe or SNr Ca2+ transient duration (time interval between two consecutive peaks) at different rotarod speeds (GPe: Pearson $r = 0.75$, SNr: $r = 0.67$) (Fig. 3H). This confirmed our initial hypothesis that dSPN GPe and SNr axons track the temporal boundaries of individual motor bouts during rotarod running on a

**Fig. 3 | dSPN GPe and SNr axons track the temporal boundaries of individual motor bouts. A** Globus pallidus externus (GPe) and substantia nigra reticulata (SNr) direct pathway striatal projection neuron (dSPN) axonal recordings using the calcium indicator jGCaMP7s. Mice are video-recorded in the rotarod set at constant speeds. Body part positions are obtained with DeepLabCut. **B** Representative trace of GPe/SNr zscored normalized fluorescence, i.e., deltaF/F (dFF), with zoom-in view (left inlet). IPI: interpeak interval. **C** Average GPe/SNr dFF traces at increasing speeds (left) and heatmaps of all individual trials at 5 rounds per minute (rpm) (3/animal) (right) showing sustained activity across the running epoch. Only the first/last 30 s of the rotarod epoch is shown (cut-off = dashed lines). **D** Representative trace of mouse lower body position on the $Y$ axis during running, showing the onset (green) and offset (black circle) of jumping bouts. dSPN GPe and SNr dFF closely track lower body Y trajectory. Dashed lines show lower (0 cm) and upper (3 cm) bounds of the rotarod. **E** Left: Probability distributions of jump IPIs. Maximal probability (vertical bar) is reached at smaller IPIs as the rotarod speed increases.

Right: Jump IPI averaged per animal significantly decreases as the rotarod speed increases (ANOVA: speed: $p < 0.01$; Tukey post-hocs: *$p < 0.05$ or **<0.01). **F** Average data aligned to the onset of individual jump bouts during running. GPe and SNr dFF increases at jump onset, showing shorter and smaller transients with increasing rotarod speed. **G** Duration of jumping bouts (peak-to-peak) and duration of GPe/SNr dFF transients (peak-to-peak) significantly decrease with rotarod speed (all: ANOVA: speed: $p < 0.01$, Tukey post-hocs: *$p < 0.05$, **<0.01, ***<0.001). **H** Duration of individual jumping bouts significantly correlate with duration of individual GPe/SNr dFF transients (Pearson $r$, all $p < 0.001$: duration: GPe: $r = 0.75$; SNr: $r = 0.67$). **I** Pearson correlation between GPe and SNr activity decreases in a task-dependent manner between rest (pre/post: $r = 0.90$) and running ($r = 0.70$) (ANOVA: epoch: $p < 0.01$, post-hoc: **$p < 0.01$). $N = 7$ mice for all experiments. Data are mean ± SEM. Exact $p$-values are given in Supplementary Dataset S2. See also Supplementary Figs. S4, S5. Source data are provided as a Source Data file.

trial-by-trial basis, consistent with previous observations at the cell body[40,48,49]. Lastly, we determined the degree of correlation between dSPN GPe and SNr axonal activity in the rotarod. Activity of GPe and SNr terminals were highly correlated with each other during rest (Pearson $r = 0.90$), similar to the open field. However, the correlation was significantly reduced ($r = 0.70$) during the running epoch (Fig. 3I). This indicated that although dSPN cell bodies send axonal copies to the GPe and SNr, differences in activity emerge during motor behavior in a task-dependent manner. Altogether these data indicate that GPe and SNr axons are concurrently, but task-dependently, activated during the running phase of a rotarod task, showing both sustained activity during the entire task and acute activation at the temporal boundaries of individual motor bouts.

### dSPN presynaptic terminal photometry in the GPe confirms task-dependent correlation between GPe and SNr terminals

dSPN calcium dynamics in the GPe could potentially be contaminated by calcium dynamics in the primary descending axon branches. In principle, collateral/terminal calcium signals should dominate since voltage-gated calcium channels are concentrated at terminals, and previous work found that calcium transients in primary axon branches are minimal as compared to calcium transients in terminals[52]. Still, to get confirmation that the activity of dSPN GPe bridging collaterals is regulated by motor tasks, we sought to perform GPe terminal-specific recordings in Drd1-cre mice expressing the calcium indicator GCaMP tethered to the presynaptic vesicle protein synaptophysin (Synaptophysin-GCaMP). Since existing Synaptophysin-GCaMP constructs harboring the calcium indicator GCaMP6s[25] have a low signal-to-noise ratio and high rate of photobleaching[53], their sensitivity is likely too low for correlational analyses.

Therefore we generated a Synaptophysin-jGCaMP8s (SyGCaMP8s) construct (Supplementary Note 1) allowing to target the next-generation jGCaMP8s calcium indicator[54] to dSPN presynaptic terminals (Fig. 4A, B, Supplementary Fig. 10). We chose jGCaMP8s due to its superior sensitivity (1 AP dFF 9.21 vs. GCaMP7s 4.95). Expression was enriched in GPe dSPN terminals, showing 6x higher fluorescent optical density in terminals vs. axons; contrasting with regular jGCaMP7s showing a 2x terminal:axon ratio (Fig. 4C). Like for jGCaMP7s (Fig. 3, Supplementary Fig. S4), we detected significant increases in calcium activity in GPe and SNr presynaptic terminals during running, evidenced by an increased baseline and AUC (Fig. 4E, F). This confirms that dSPN GPe terminals are engaged during motor tasks. Importantly, like for jGCaMP7s, activity of GPe and SNr terminals were highly correlated with each other during rest (Pearson $r = 0.69$), and the correlation was significantly reduced ($r = 0.47$) during running (Fig. 4G). Since Pearson $r$ values obtained with jGCaMP7s were 20–30% higher than with synaptophysin-jGCaMP8s this suggests that 20–30% of the correlation in the jCCaMP7s could be due to $Ca^{2+}$ signals in axons. It is likely though that even with the terminal-targeted SyGCaMP8s there may still be a

small contribution of axonal signal. Moreover, because jGCaMP8s has faster on and off kinetics than jGCaMP7s (half rise-time: 21 vs. 67 ms; half-decay time: 52 vs. 81 ms[54]. Note: SyGCaMP8s kinetics may slightly differ vs. jGCaMP8s. See also Supplementary Fig. S11), differences in the sensor kinetics could contribute to the differences in the degree of correlations. Regardless of the source of variation, we find that results obtained with both sensors concur, namely that correlated activity is higher in the rotarod off vs. on condition. This confirms our finding that dSPN cell bodies send shared information to the terminals in GPe and SNr, but that there are additional local factors in the GPe/SNr that regulate terminal activity in a task-dependent manner.

### dSPN bridging collaterals in the GPe are necessary for motor function

Since dSPN GPe terminals encode locomotion and rotarod motor variables, we next asked if they are necessary for normal locomotion and motor function. Selectively manipulating dSPN bridging collateral activity is not trivial. Using classical excitatory opsins such as ChR2 is precluded since they would affect anterograde and retrograde action potential propagation in dSPN passing fibers in the GPe (Fig. 1B). Moreover many inhibitory opsins have off-target effects[25]. We therefore used the inhibitory DREADD hM4D, as it was previously shown to inhibit synaptic release with minimal effects on action potentials in axons[55] and used to target dSPN collaterals in the ventral pallidum during cocaine seeking[56]. Drd1-cre mice were bilaterally injected with a flexed AAV expressing hM4D or mCherry into the dStr and implanted with GPe cannulas for infusion of the DREADD agonist clozapine-N-oxide (CNO) (Fig. 5A, B, Supplementary Fig. S10). Local infusion allows to bypass potential liver metabolization into clozapine[57] and selectively inhibits GPe terminals. First, we verified that locally infused radioactive [³H]-CNO (300 nL; 7 μCi/mL) stayed restricted in the GPe (Fig. 5C). We then infused the same volume (300 nL) of CNO (at 1 mM) into the GPe 20–30 min before the behavioral analysis. We found that chemogenetic inhibition of bridging collaterals reduced locomotor speed in the open field (Fig. 5D), shown by a significant reduction in mouse speed in hM4D but not in mCherry control mice. The same manipulation also impaired rotarod motor performance, shown by a non-significant decrease in latency to fall and a significant increased number of falls in hM4D but not mCherry mice (Fig. 5E). These data suggest that bridging collaterals support and are necessary for motor control.

Of note, in this experiment dSPN passing axons fibers in the GPe going to the SNr are physically exposed to the locally infused CNO. We therefore set to verify that our hM4D results could not be explained by unspecific effects in the SNr. Indeed, although previous work showed that CNO + hM4D inhibits synaptic release with minimal effects on action potentials in axons[55], this was done in cortical neurons, which may have different biophysical properties than dSPNs. Drd1-cre mice were injected with a mix of flexed AAVs

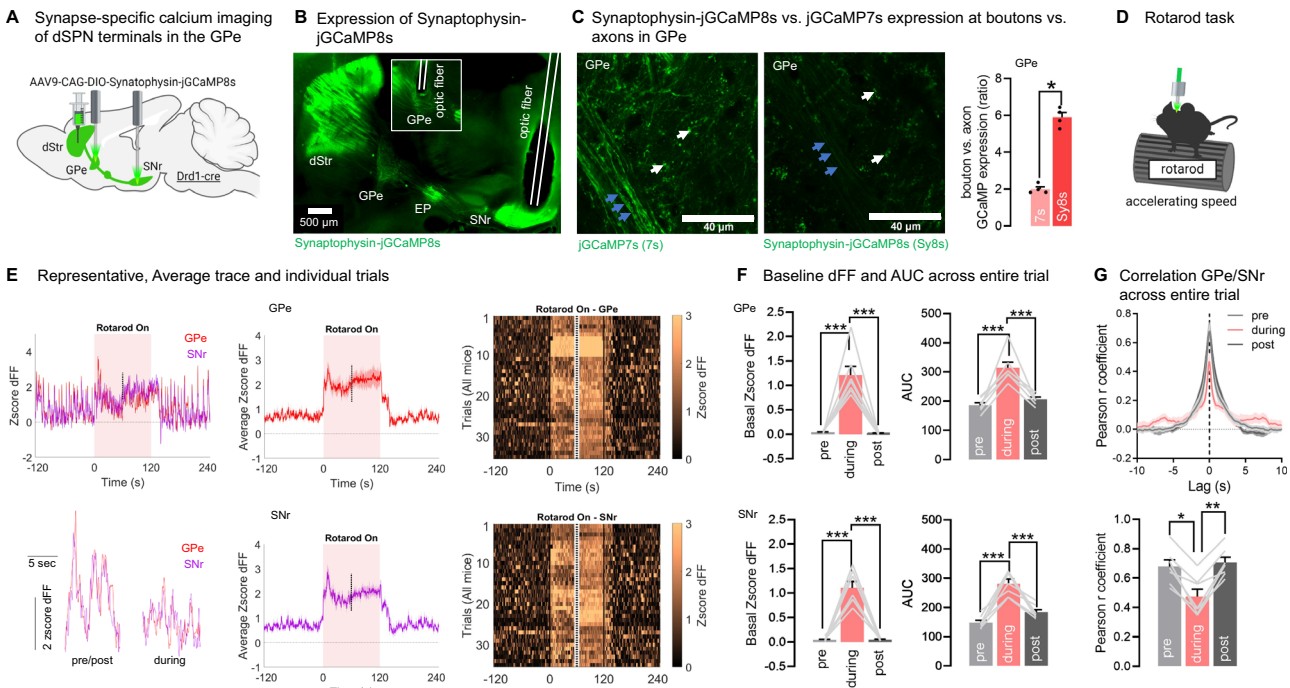

**Fig. 4 | dSPN presynaptic terminal photometry in the GPe and SNr confirms task-dependent correlation in a rotarod motor task. A** Strategy for terminal-specific calcium recordings with fiber photometry using the presynapse-targeted calcium sensor Synaptophysin-jGCaMP8s for direct pathway striatal projection neuron (dSPN) terminals in the globus pallidus externus (GPe) and substantia nigra reticulata (SNr) **B** Synaptophysin-jGCaMP8s expression in the dorsomedial section of the dorsal striatum (dStr) with optic fibers targeting the GPe (inlet from other section) and SNr, representative from $N = 7$. EP entopeduncular nucleus. **C** Left: Synaptophysin-jGCaMP8s (Sy8s) is expressed in terminals (white arrows) but poorly in axons in the GPe at 10–14 days post injection. As a comparison, untargeted jGCaMP7s is detected in terminals (white arrows) and axons (blue arrows) in the GPe. Right: Quantification of optical density in boutons vs. axons in the GPe shows sixfold bouton enrichment in Synaptophysin-jGCaMP8s and twofold in jGCaMP7s brains (ANOVA: epoch: $p < 0.001$, Tukey post-hocs: *$p < 0.05$). Quantification made in unstained fixed brains (native fluorescence) to avoid potential antibody amplification artefacts ($N = 4$ mice). **D** Mice are tested in the rotarod at accelerating speeds 10–14 days post injection. **E** Representative trace of GPe/SNr zscored normalized fluorescence, i.e., deltaF/F (dFF) and zoom-in inlet (left) for an individual animal, average trace of all mice (middle) and heatmaps of all individual trials (right) showing terminal-specific dSPN GPe and SNr activity in the rotarod task. Only the first and last 30 s of the rotarod epoch is shown (cut-off = dashed lines). **F** GPe and SNr terminal activity shows a significant increase in baseline and area under the curve (AUC) in the run epoch (during) vs rest (pre/post) (all: ANOVA: epoch: $p < 0.001$; Tukey post-hoc: all: ***$p < 0.001$). **G** Pearson correlation between GPe and SNr activity decreases in a task-dependent manner between rest (pre/post: $r = 0.69$) and running ($r = 0.47$) (ANOVA: epoch: $p < 0.01$, Tukey post-hocs: *$p < 0.05$, **$p < 0.01$). $N = 7$ mice for all photometry experiments. Exact $p$-values are given in Supplementary Dataset S2. Data are mean ± SEM. Source data are provided as a Source Data file.

expressing ChR2 and hM4D into the DMS. We used our previously validated setup[6] to record single-unit responses in the GPe and SNr after acute optogenetic stimulation of dSPN somas at increasing durations (0, 250, 500, 1000 ms) in anesthetized mice. We also locally infused Saline or CNO (300 nL, 1 mM) above the GPe 25 min before recording to inhibit synaptic release at dSPN GPe terminals (Fig. 6A, B, Supplementary Fig. S10). Consistent with previous work[6,17], in Saline control mice dSPN opto-stimulation led to an inhibition of spike firing frequency in the GPe (Fig. 6C) and SNr (Fig. 6E). An average of 55% (considering normalized spike frequencies, see Fig. 6G) or 33% (considering Z-scores, see Fig. S6C) of GPe neurons were inhibited, which could be due to a mix of monosynaptic and polysynaptic effects since dSPNs are thought to target Npas1 (~30% of GPe or cells) and ChAT cells (5%)[15–19]. The dSPN opto-induced inhibition of GPe spike firing was blunted when dSPN GPe terminals were chemogenetically inhibited via local GPe CNO infusion (Fig. 6D). This confirmed that local GPe CNO infusion in hM4D-expressing Drd1-cre mice (Fig. 5) inhibits synaptic release at dSPN GPe terminals, in line with the established role of hM4D as a presynaptic release inhibitor[55]. Importantly, local CNO infusion into the GPe did not affect opto-induced inhibition of SNr spike firing activity (Fig. 6F). Upon quantification, we found that local GPe CNO infusion significantly reduced the number of inhibited units in the GPe (Fig. 6G), but not in the SNr (Fig. 6H). Similarly, local GPe CNO

infusion significantly blunted the opto-induced inhibition of spike frequency in GPe units (Fig. 6I; see Fig. 6J for the first 50 ms), but not of SNr units (Fig. 6K; see Fig. 6L for the first 50 ms). Since spike frequency in the SNr appears to be affected by CNO early during the inhibition we restricted the analysis to only the first 50 ms of optogenetic stimulation (Fig. 6J, L) but did not detect any effect of CNO in the SNr. We confirmed these data using a Z-score based analysis which provided a more stringent criterion for identifying units whose activity was decreased by the optogenetic inhibition (Supplementary Fig. S6C–H). Although the Z-score appears slightly decreased in the SNr with CNO, we could not detect any significant effects, either by looking at the full stimulation window or the first 50 ms (Supplementary Fig. S6D, F, H). We also identified a low number (0–8) of excited units, but the number was too low for a statistical comparison between the saline and CNO groups (Supplementary Fig. S7). Last, while baseline firing in the SNr appears to be lower in CNO injected mice (possibly due to polysynaptic effects), this effect was not significant (Supplementary Fig. 6A, B). These in vivo physiology results indicate that local infusion of CNO into the GPe inhibits synaptic release at local dSPN GPe terminals but does not significantly affect action potential propagation in descending dSPN axons going to the SNr. However, despite these negative findings we cannot entirely exclude the existence of possible activity changes (mono or polysynaptic) in the SNr after GPe

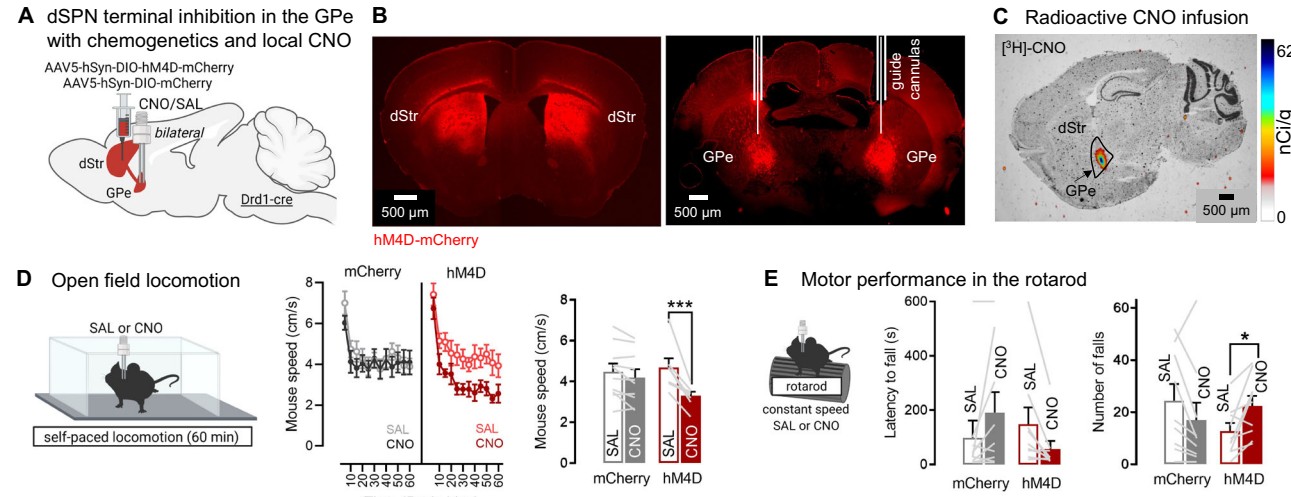

**Fig. 5 | dSPN bridging collaterals in the GPe support motor function as revealed with chemogenetic inhibition. A** Strategy for chemogenetic inhibition of direct pathway striatal projection neuron (dSPN) terminals in the globus pallidus externus (GPe) using GPe infusion of clozapine-N-oxide (CNO) and the inhibitory designer receptor hM4D. dStr: dorsal striatum. **B** Fluid cannulas target dSPN terminals in the GPe expressing hM4D-mCherry, representative from $N = 10$ mCherry, 9 hM4D. **C** Representative image from $N = 3$ of local radioactive [³H]-labeled clozapine-N-oxide (CNO) infusion confirming the drug can stay restricted in the GPe at this volume (300nL). **D** Left: Mice are tested in the open field after infusion of Saline (SAL) or CNO. Right: Chemogenetic inhibition of dSPN GPe terminals significantly reduces locomotion speed (ANOVA: virus x drug $p < 0.001$; Sidak post-hoc: SAL vs CNO: hM4D $p < 0.001$, mCherry $p = 0.32$), $N = 10$ mCherry, 7 hM4D. **E** Left: Mice are tested in the rotarod at constant speed after SAL/CNO infusion. Mice are allowed to return to the rotarod if they fall. Right: Chemogenetic inhibition of dSPN GPe terminals significantly increases the number of falls (ANOVA: virus x drug $p < 0.05$; Sidak post-hoc: SAL vs CNO: hM4D $p < 0.05$, mCherry $p = 0.12$). $N = 9$ mCherry, 9 hM4D. Exact $p$-values are given in Supplementary Dataset S2. Data are mean ± SEM. Source data are provided as a Source Data file.

CNO infusion. Together with the behavioral data, this supports the notion that dSPN GPe terminals are necessary for normal locomotion and motor control.

To confirm these findings and gain higher temporal resolution, we next used the recently developed Gi/o mosquito rhodopsin eOPN3 shown to selectively inhibit synaptic release while maintaining action potential fidelity in axons[58]. Drd1-cre mice were bilaterally injected with a flexed AAV expressing eOPN3 or GFP into the DMS and implanted with GPe optic fibers (Fig. 7A, B, Supplementary Fig. S10). As expected, optogenetic inhibition of dSPN GPe terminals impaired rotarod motor performance, as shown by a significant decreased latency to fall detected in eOPN3 but not GFP controls (Fig. 7C). We next performed a closed-loop open field task to inhibit bridging collaterals during ongoing locomotion: here the optogenetic light was activated only when mice were actively locomoting (see Methods) and mouse speed was compared in laser-on vs. laser-off epochs. We found that closed-loop optogenetic inhibition of dSPN GPe terminals reduced ongoing locomotion speed, as shown by a significant reduction in mouse speed detected in eOPN3 but not GFP mice (Fig. 7D, Supplementary Video S3). We then classified behaviors into motor states to dissect the fine motor patterns induced by dSPN GPe inhibition. Trajectories of mouse body parts (obtained with DeepLabCut) were used to classify frames into three categories: locomotion, motionless and non-locomotor movements (this includes but are not restricted to head movements, rearing and grooming). We found that opto-inhibition of dSPN GPe terminals promoted motor states consistent with decreased motion but not with behavioral arrest, as shown by our observations of a significantly increased time spent in non-locomotor movements and a decreased time spent locomoting, but no change in time spent motionless (Fig. 7E, F). There were no differences in time spent in center vs. periphery zones, suggesting no effects on anxiety (Supplementary Fig. S8A). Since dSPN inhibition blunts locomotion and rotarod motor performance, altogether, these results show that dSPN GPe terminals normally support locomotion and motor performance in the rotarod, suggesting they act as a second Go pathway.

## dSPN axons inhibit ongoing motor-related calcium dynamics in their GPe Npas1 target neurons

What could be the circuit mechanisms by which dSPN GPe terminals support motor function in the GPe? The GPe can be divided into two principal neuron classes, arkypallidal neurons (which all express FoxP2, most of which express Npas1; and which heavily project to the striatum) and prototypical neurons (which express Nkx2.1, most of which express parvalbumin, PV; and project to the midbrain)[16,20,22,23,59–61]. We here hypothesized that dSPN axons support motor function by functionally inhibiting ongoing motor-related activity in arkypallidal neurons[22,23]. Indeed, recent work used Npas1-cre and PV-cre mice to label arkypallidal and prototypical neurons, respectively: they found that although PV+ neurons receive limited dSPN input and their activation promotes locomotion, Npas1+ neurons receive strong dSPN input and their activation suppresses locomotion[16,62,63]. Similarly in the in vivo anesthetized state[17,19], dSPNs were shown to strongly inhibit FoxP2+ neurons but only weakly inhibit Nkx2.1+ neurons. Therefore in a first step we set to determine whether the dSPN to arkypallidal connection is active during ongoing motor behavior. Indeed it is important to verify this since recent work showed that active circuits in the GPe cannot always be predicted from in vivo physiology experiments done in the anesthetized state due to the presence of dense multisynaptic inhibitory circuits which can override monosynaptic connections to arkypallidal cells[15]. We here used Drd1-cre mice crossed to Npas1-cre mice. In mice or rats 55–70% of Npas1 neurons are considered arkypllidal (the other 30–45% project to cortex, reticular thalamus and midbrain instead of striatum)[20,59–62]; thus FoxP2-cre mice would be more selective for arkypallidal cells than Npas1-cre mice. However, FoxP2 is also expressed in the striatum which would interfere with our dSPN manipulation, while Npas1-cre selectively expresses in the GPe (see Fig. 1 in ref. 64).

Drd1-cre;Npas1-cre mice were injected with a flexed AAV expressing ChrimsonR or mCherry into the DMS to target dSPNs and a flexed AAV expressing GCaMP6s into the GPe to target Npas1 cells. An optic fiber was implanted above the GPe to optogenetically stimulate dSPN

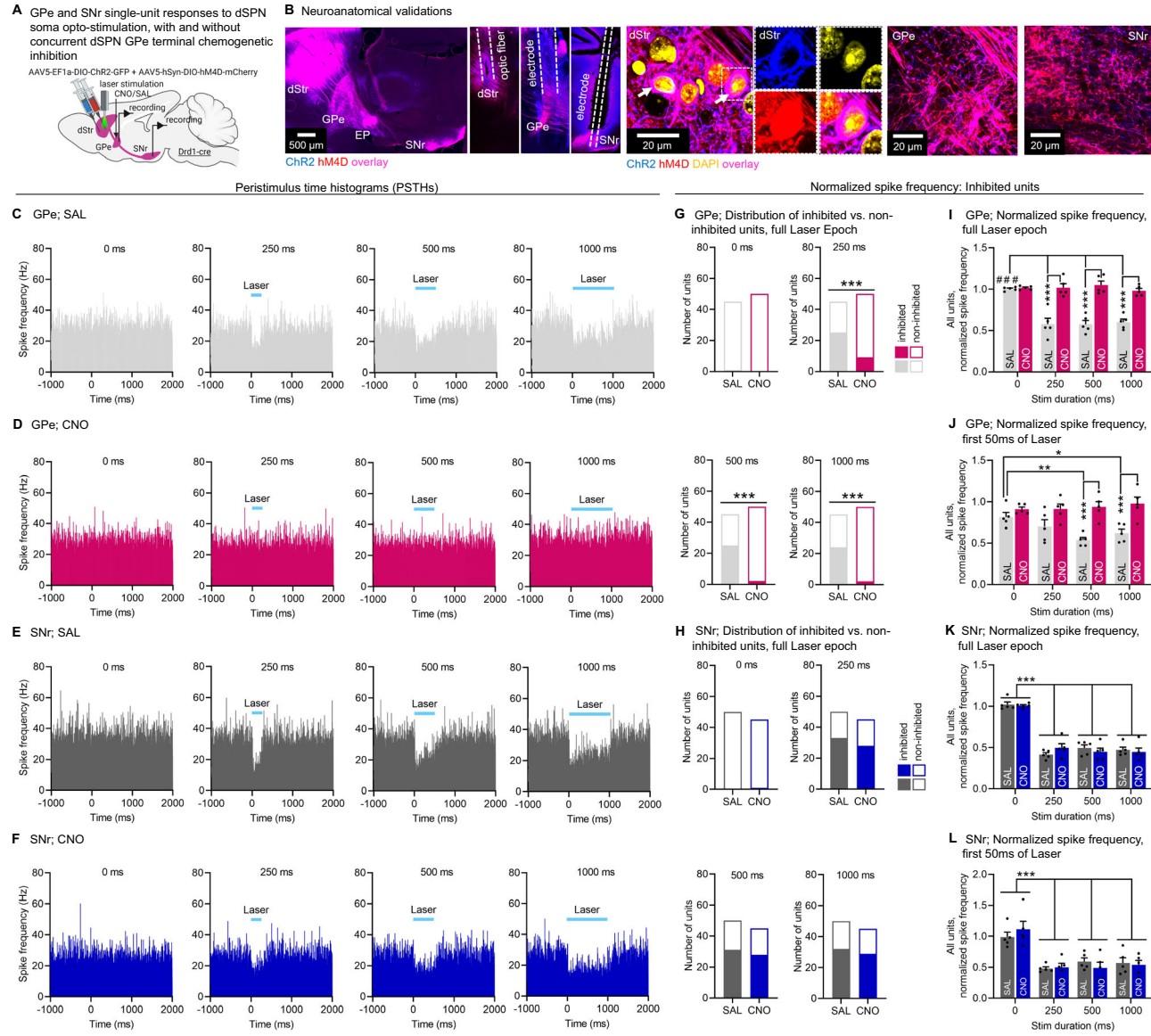

**Fig. 6 | Confirmation that chemogenetic manipulation of dSPN GPe terminals does not affect activity in the SNr. A** Strategy for in vivo recordings of globus pallidus externus (GPe) and substantia nigra reticulata (SNr) units following opto-stimulation of direct pathway striatal projection neuron (dSPN) somas with ChR2 and chemogenetic inhibition (with hM4D) of dSPN GPe terminals using local Saline (SAL) or clozapine-N-oxide (CNO, 1 mM/300 nL) infusion **B** Expression of hM4D-mCherry (red) and ChR2-YFP (blue) and their colocalization (white arrow) in dSPN somas in the dStr (see optic fiber tracks) and terminals in the GPe/SNr (see electrode tracks). **C, D** Peristimulus time-histograms (PSTHs) showing mean spike frequency of all recorded GPe neurons before, during, and after laser-stimulation (1 ms bins) in all animals with GPe SAL (**C**) or CNO (**D**) treatment. **E, F** Same as **C, D** for the SNr. **G** Proportion of GPe units for which basal firing activity is significantly decreased (inhibited units) or not (non-inhibited units) after laser-stimulation at different durations calculated in the full laser epoch (Fisher's test SAL vs. CNO at 250, 500, 1000 ms: ***$p < 0.001$; at 0 ms $p = 0.48$). **H** Same as **G** for the SNr (Fisher's test at all stim durations: SAL vs CNO: $p = 0.48$-0.99). **I** Normalized

spike frequency in the GPe calculated in the full laser epoch normalized to the 1000 ms pre-stimulation period for all units (ANOVA: stim duration x drug $p < 0.001$; Sidak post-hocs SAL vs. CNO at 250, 500, 1000 ms: all ***$p < 0.001$; Sidak post-hocs 0 ms vs. other durations: all SAL: #$p < 0.001$, all CNO: $p = 0.8$–1.0). **J** Same as **I** calculated in the first 50 ms of the laser epoch (ANOVA: stim duration x drug $p = 0.0313$; Sidak post-hocs SAL vs. CNO at 250 ms $p = 0.0586$, 500 and 1000 ms ***$p < 0.001$; Sidak post-hocs: SAL 0 vs. 250 ms $p = 0.34$, 0 vs. 500 ms: **$p = 0.0032$, 0 vs 1000 ms *$p = 0.0464$, CNO: all $p = 0.8$–1.0). **K.** Normalized spike frequency rate in the SNr in the full laser epoch (ANOVA: stim duration x drug $p = 0.16$; drug $p = 0.99$, main effect of stim duration: ***$p < 0.001$; Sidak post-hoc all mice pooled: 0 ms vs. other durations: all ***$p < 0.001$). **L** Same as **K** calculated in the first 50 ms (ANOVA: stim duration x drug $p = 0.47$, drug $p = 0.96$, main effect of stim duration: ***$p < 0.001$; Sidak post-hoc all mice pooled: 0 ms vs. other durations: all ***$p < 0.001$). $N = 5$ GPe;SAL, 5 GPe;CNO, 5 SNr;SAL, 5 SNr;CNO throughout. Exact $p$-values given in Supplementary Dataset S2. Data are mean ± SEM. See also Supplementary Figs. S6, S7. Source data are provided as a Source Data file.

axons in the vicinity of GCaMP+-Npas1 neurons (Fig. 8A, B; Supplementary Fig. S10). Behavior was quantified using body positions obtained from DeepLabCut. We used unilateral stimulation of dSPN GPe axons to minimize effects on behavior, and disentangle them from effects on calcium activity. In a first approach, we asked if dSPN axons inhibit ongoing Npas1 activity. Here closed-loop dSPN axon stimulation was triggered when Npas1 activity reached a maxima (see

Methods, Fig. 8C), based on previous work showing that Npas1 activity is high at locomotor onset[59]. Due to this approach Npas1 activity peaks at the time point when the closed-loop stimulation is initiated (Fig. 8D). We found that dSPN axon stimulation for short or long durations (3 or 10 s; 20 Hz) led to the inhibition of Npas1 activity in a graded manner (stronger inhibition with 2 mW vs. 0.5 mW) and observed in all 6 ChrimsonR mice recorded (Fig. 8D, Supplementary Fig. S9A,

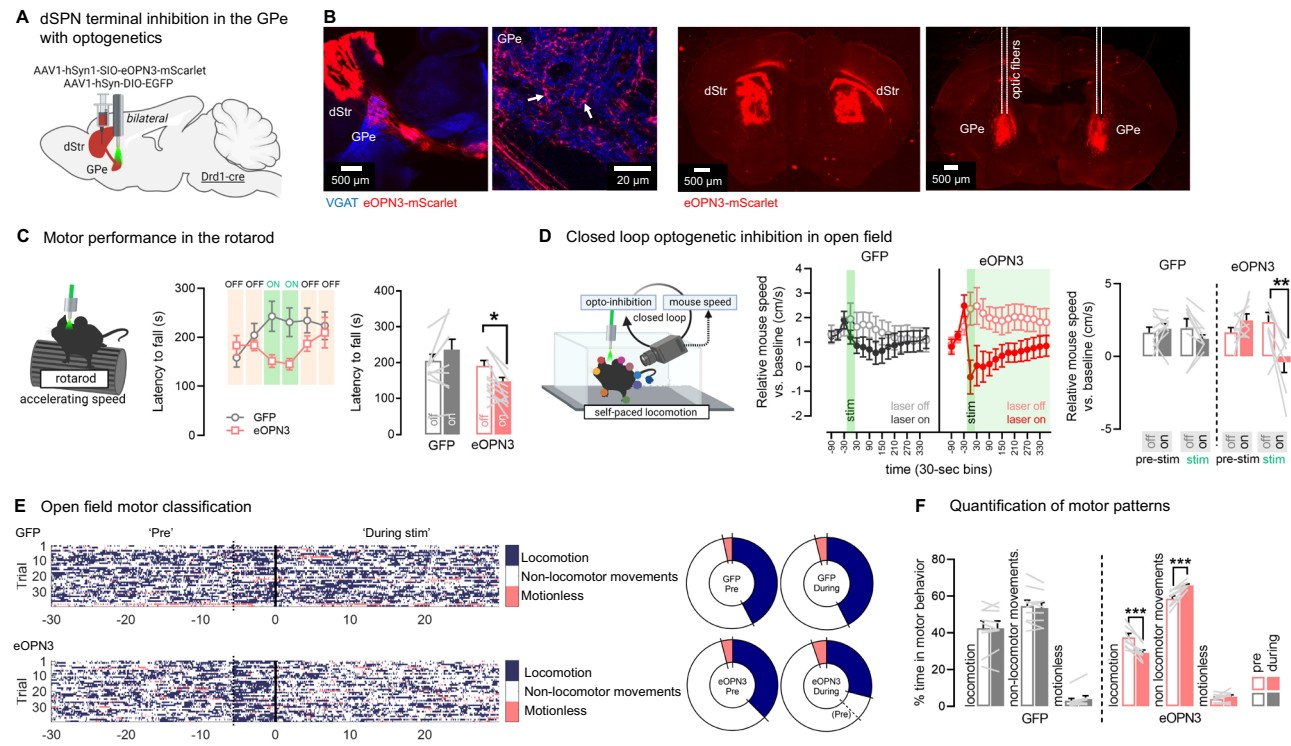

**Fig. 7 | dSPN bridging collaterals in the GPe support motor function as revealed with optogenetic inhibition. A** Strategy for optogenetic inhibition of direct pathway striatal projection neuron (dSPN) terminals in the globus pallidus externus (GPe) using the inhibitory opsin eOPN3. dStr: dorsal striatum. **B** Left: eOPN3-mScarlet (red) colocalizes with the presynapse marker VGAT (blue) in the GPe, appearing pink (white arrows). Right: Optic fibers target dSPN eOPN3-mScarlet+ terminals in the GPe. **C** Left: Optogenetic inhibition during rotarod trials at accelerating speed. Middle: Optogenetic inhibition of dSPN GPe terminals reduces latency to fall. Right: Summary data showing significance (ANOVA: Virus x Laser $p < 0.01$; post-hoc: on vs off: eOPN3 *$p < 0.05$, GFP $p = 0.10$). Mice were only allowed to fall once. $N = 10$ eOPN3, 9 GFP. **D** Left: Optogenetic inhibition triggered in a closed-loop during ongoing locomotion (see Methods). ITI intertrial interval. Body part positions obtained with DeepLabCut. Middle: Optogenetic inhibition (30 s) of dSPN GPe terminals reduces mouse speed, which recovers after 6 min (consistent with[58]). Right: Summary data showing significant reductions in mouse speed in the stimulation (stim) epoch (green) vs. the 30 s preceding (pre-stim, black), but only when the laser was turned on (vs. off) (ANOVA: Virus x Laser x Epoch $p < 0.05$; Sidak post-hoc: on vs off in the post-epoch: eOPN3 **$p < 0.01$, GFP $p = 0.96$), $N = 8$ eOPN3, 9 GFP. **E** Left: Heatmaps showing behavioral classification of videoframes (all mice) into locomotion (body center speed >4.5 cm/s), motionless (speed of all body parts ≤0.8 cm/s) or other non-locomotor movements (does not fulfil locomotion or motionless criteria). Non-locomotor movements include but are not restricted to head movements, rearing, grooming and other fine movements. Note the mild locomotion increase 5 s before laser onset (dashed line) due to the closed-loop. Right: %frames in each motor classification, showing decreased locomotion and increased non-locomotor movements during dSPN GPe inhibition, $N = 8$ eOPN3, 9 GFP. **F** dSPN GPe inhibition significantly (Mixed ANOVA: virus x epoch x motor-classification: $p < 0.001$) reduces %time spent locomoting (Sidak post-hocs: eOPN3 ***$p < 0.001$, GFP $p = 0.99$) and increases %time spent in non-locomotor movements (Sidak post-hocs: eOPN3 ***$p < 0.001$, GFP $p = 0.61$), $N = 8$ eOPN3, 9 GFP. Data are mean ± SEM. Exact $p$-values given in Supplementary Dataset S2. See also Supplementary Fig. S8. Source data are provided as a Source Data file.

Supplementary Video S4), but not in mCherry controls (Supplementary Fig. S9C). Stimulating dSPN axons at 20 Hz but not 10 Hz significantly inhibited Npas1 neurons, suggesting Npas1 inhibition occurs only when activation of dSPNs reaches a certain threshold (Fig. 8F). Importantly, these neural effects were decorrelated from effects on behavior. Indeed, as expected, unilateral stimulation did not affect mouse speed (Fig. 8E, G). Significant increases in rotational behavior emerged after 10 but not 3 s unilateral stimulation protocols (Supplementary Fig. S9B). This suggested that the effects of dSPN stimulation on Npas1 ativity could not be solely explained by changes in mouse behavior. In a second approach, we asked if dSPN axons inhibit motor-related Npas1 signals. Here closed-loop dSPN unilateral axon stimulation was triggered when the mouse was actively locomoting (see Methods, Fig. 8H). Stimulation was done at ultra-low ChrimsonR-LED power (0.2 mW), which had no significant effects on mouse speed (Fig. 8I) or rotations (Supplementary Fig. S9B). As expected[59], before stimulation Npas1 dFF activity increased concurrently with mouse speed (Fig. 8I, J). We found that dSPN axon stimulation was sufficient to inhibit motor-related Npas1 calcium activity, even at this low optogenetic power (Fig. 8J). Finally, we also verified the absence of crosstalk between the LEDs required to activate ChrimsonR and GCaMP,

respectively (shown in Supplementary Fig. S9B, C). Altogether these findings show that dSPNs significantly impact motor-related signals in the GPe by inhibiting their Npas1 target neurons in awake locomoting mice, emphasizing the physiological relevance of the dSPN-Npas1 circuit for behavior.

## GPe Npas1 but not ChAT neurons mediate the effects of bridging collaterals on motor function

Our data suggest a mechanism by which dSPN bridging collaterals support motor function by inhibiting Npas1 neurons in the GPe during ongoing behavior. This would align with past research showing that Npas1 neurons are locomotor-suppressing[15,63]. Since bridging collateral inhibition reduces locomotion speed and impairs rotarod motor performance, we wondered whether disinhibition of Npas1 neurons could recapitulate these phenotypes. Npas1-cre mice were bilaterally injected with a flexed AAV expressing ChR2 or YFP into the GPe and implanted with GPe optic fibers (Fig. 9A, B, Supplementary Fig. S10). We stimulated Npas1 neurons at 20 Hz around their firing frequency during ongoing locomotion[59]. Optogenetic stimulation of Npas1 neurons inhibited locomotion speed, as shown by a significant reduction in mouse speed in ChR2 mice but not YFP controls (Fig. 9C),

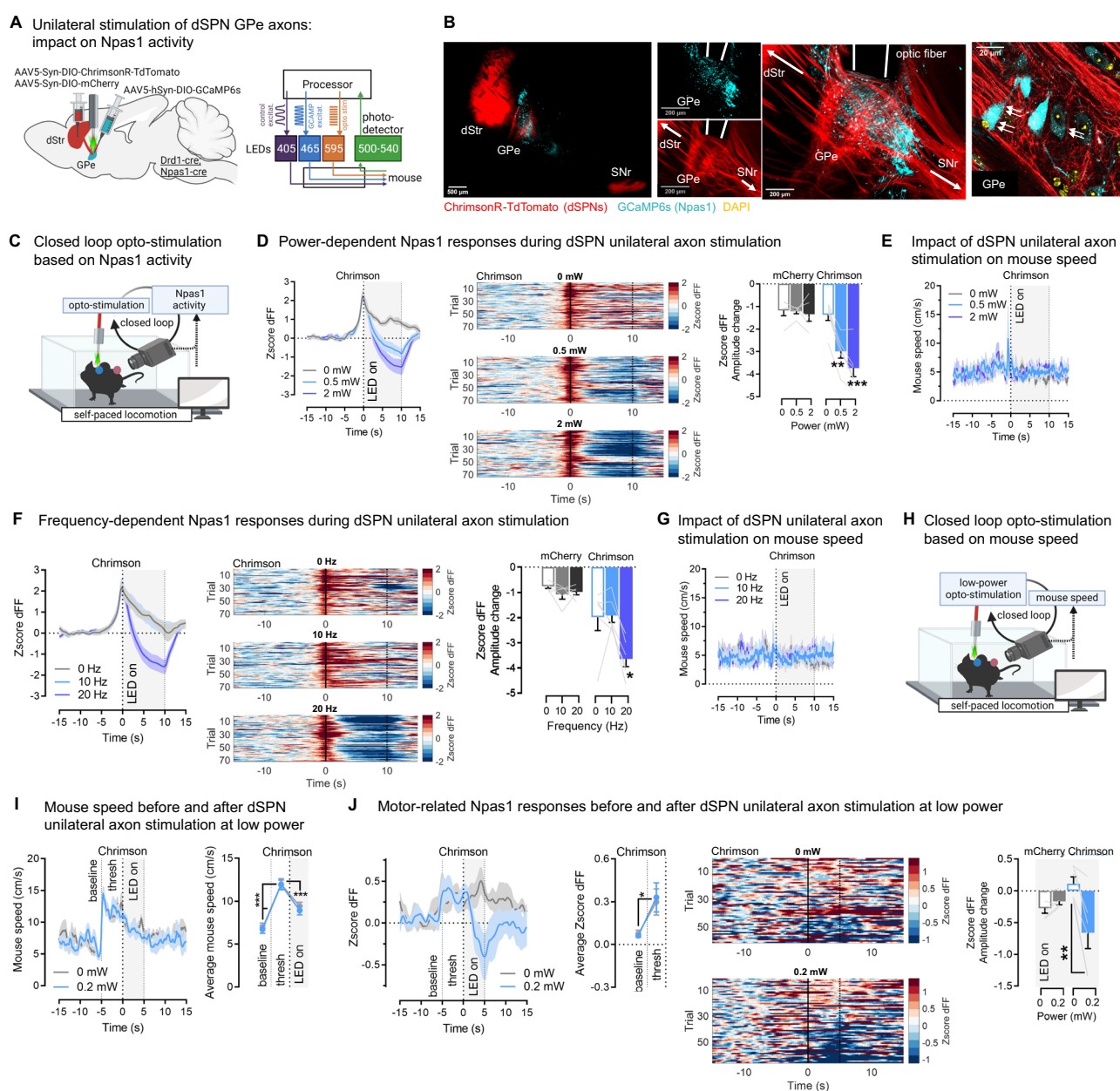

**Fig. 8 | dSPN axons inhibit ongoing motor-related calcium dynamics in their GPe Npas1 target neurons. A** Left: Optogenetic stimulation of direct pathway striatal projection neuron (dSPN) axons in the globus pallidus externus (GPe) using the opsin ChrimsonR; simultaneous recording of Npas1 activity using the calcium indicator GCaMP6s. dStr dorsal striatum. Right: All-optical setup. **B** ChrimsonR-TdTomato+ dSPN terminals in the GPe (red) apposed to (white arrows) GCaMP6s +Npas1 somas (cyan). Optic fibers in the same region. Representative from $N = 6$ mice. **C** Opto-stimulation is triggered in a closed-loop when Npas1 dFF surpasses a defined threshold (see Methods). **D** 10 s, 20 Hz stimulation of dSPN GPe axons leads to a power-dependent reduction in Npas1 activity. Left: Average traces, Middle: Heatmaps of all trials, Right: Amplitude change in Npas1 dFF in the opto-window (ANOVA: virus x power $p < 0.001$; Sidak post-hoc: Chrimson **$p < 0.01$, ***$<0.001$, mCherry $p > 0.8$). **E** As expected: no effect of unilateral stimulation on mouse speed (ANOVA: power $p = 0.50$). **F** 10 s, 2 mW stimulation of dSPN GPe axons leads to a frequency-dependent reduction in Npas1 activity. Left: Average traces, Middle: Heatmaps of all trials, Right: Amplitude change in Npas1 dFF in the opto-window

(ANOVA: virus x power $p < 0.01$; Sidak post-hocs: Chrimson: 0 vs 20 Hz *$p < 0.05$, 0 vs 10 Hz $p = 0.99$, mCherry $p = 0.28$ and $0.51$). **G** No effect of stimulation on mouse speed (ANOVA: power $p = 0.56$). **H** Opto-stimulation triggered in a closed-loop during ongoing locomotion when mouse speed reaches a defined threshold (see Methods). **I** As expected, mouse speed increases before the opto-trigger (baseline vs. threshold (thresh) and stim periods), but is not affected by opto-stimulation (0 vs. 0.2 mW) (ANOVA: epoch $p < 0.001$, epoch x LED $p = 0.79$; Sidak post-hocs all ***$p < 0.001$). **J** 5 s, 20 Hz stimulation of dSPN GPe axons at ultra-low power (0.2 mW) reduces motor-related Npas1 activity. Left: Average and summary data showing a significant increase in dFF before the opto-trigger (ANOVA: epoch $p < 0.05$). Middle: Heatmaps of all trials, Right: Amplitude change in Npas1 dFF in the opto-window (ANOVA: virus x LED $p < 0.01$; Sidak post-hocs: Chrimson **$p < 0.01$, mCherry $p = 0.84$). Heatmaps: straight/dashed line = LED onset/offset. $N = 5$ mCherry, $N = 6$ ChrimsonR throughout. Data are mean ± SEM. Exact $p$-values are given in Supplementary Dataset S2. See also Supplementary Fig. S9. Source data are provided as a Source Data file.

confirming the locomotor-suppressing role of Npas1 neurons[15,63]. Interestingly, optogenetic stimulation of Npas1 neurons mildly but significantly impaired rotarod motor performance, as shown by a significant decreased latency to fall in ChR2 mice but not YFP controls

(Fig. 9D). Thus, Npas1 neurons are the likely mediators of both the open field locomotion and rotarod motor performance phenotypes observed with bridging collateral manipulation. Previous work also showed that dSPN bridging collaterals provide monosynaptic

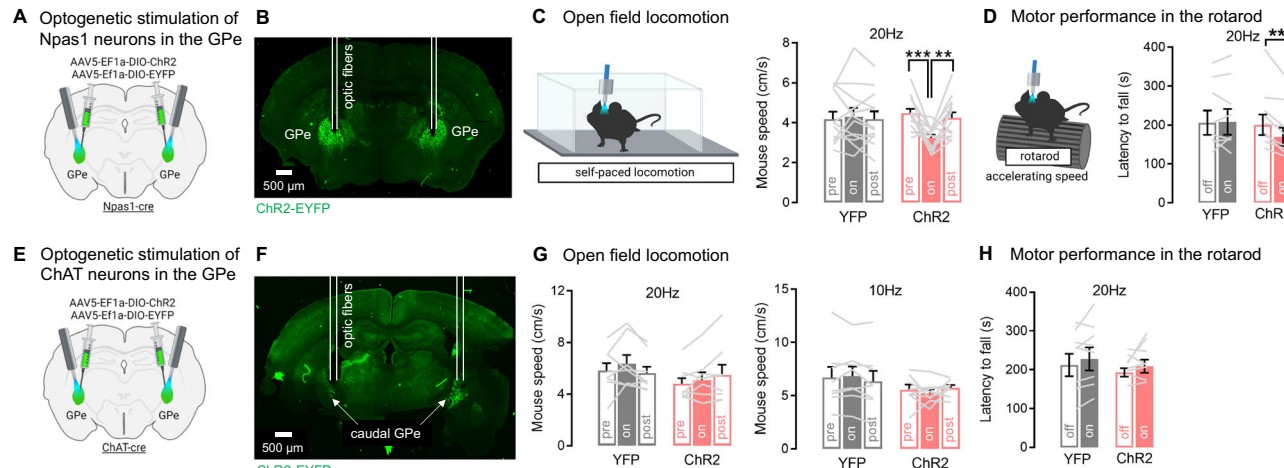

**Fig. 9 | GPe Npas1 but not ChAT neurons mediate the effects of bridging collaterals on motor function. A** Optogenetic stimulation of globus pallidus externus (GPe) Npas1+ neurons using the optogenetic activator ChR2 **B** Optic fibers target Npas1 ChR2-YFP+ neurons in the GPe. **C** Left: 10 trials optogenetic stimulation during open field locomotion. Right: 20 Hz stimulation of Npas1 neurons significantly reduces mouse speed (ANOVA: virus x epoch $p < 0.01$; Sidak post-hocs ChR2 **$p < 0.01$, ***$<0.001$, GFP $p > 0.9$), $N = 12$ ChR2, 13 YFP. **D** Left: Optogenetic stimulation (20 Hz) during rotarod trials at accelerating speed. Right: Stimulation of Npas1 neurons significantly reduces latency to fall (ANOVA: virus x laser $p < 0.05$;

Sidak post-hocs: ChR2 **$p < 0.01$, GFP $p = 0.97$), $N = 8$ ChR2, 9 YFP. **E** Optogenetic stimulation of GPe choline acetyltransferase (ChAT)+ cholinergic neurons. **F** Optic fibers target ChAT ChR2-YFP+ neurons in the caudal GPe. **G** Neither 10 Hz (ANOVA: virus x epoch $p = 0.11$) or 20 Hz ($p = 0.06$) ChAT stimulation affects mouse speed, $N = 8$ ChR2, 8 YFP. **H** 20 Hz stimulation of ChAT neurons does not affect latency to fall in the rotarod (ANOVA: virus x laser $p = 0.63$), $N = 8$ ChR2, 8 YFP. Data are mean ± SEM. Exact $p$-values are given in Supplementary Dataset S2. Source data are provided as a Source Data file.

connections to a small population of ChAT neurons located at the caudal-ventral GPe border near the basal forebrain (BF)[65]. The role of these neurons in behavior is, however, unknown. We determined whether GPe ChAT neurons are also motor-suppressing and could mediate the motor effects of bridging collaterals. ChAT-cre mice were bilaterally injected with a flexed AAV expressing ChR2 or YFP into the caudal-ventral GPe and implanted with GPe optic fibers (Fig. 9E, F, Supplementary Fig. S10). We stimulated ChAT neurons at 10 and 20 Hz in the open field, firing frequencies that BF ChAT neurons reach in vivo[66]. Optogenetic stimulation of ChAT neurons did not affect locomotion speed in the open field (Fig. 9G) or rotarod motor performance (Fig. 9H). Since GPe ChAT neurons project to the cortex, reticular thalamus, and amygdala[65], they may mediate other behavioral functions of dSPNs and warrant further study. Altogether, these data indicate that dSPN bridging collaterals support motor function via GABAergic inhibition of their Npas1 targets in the GPe, rather than via their ChAT target cells.

## Discussion

Within the vertebrate brain, motor behavior is controlled not by one descending projection but by the coordinated activity of interconnected brain circuits. However the classical model of the basal ganglia (BG), one of the central regulators of motor function, depicts a brain circuit composed of segregated descending projections sending unidirectional axons to individual brain regions[1]. This classical BG model does not account for old and more recent anatomical tracing studies which find that several BG nuclei possess arborized axons which collateralize across one or multiple BG regions on their way to their final target[8,10]. Like in other neural systems[10,67–69], these axon collateral populations could help to coordinate information flow and motor behavior by sending axonal copies of motor signals to different brain regions. Until now, however, the in vivo dynamics and behavioral functions of axon collateral populations in the BG had not been investigated.

Here we address this question focusing on striatal dSPNs by selectively recording or inhibiting dSPN axon collaterals in the GPe. We find that dSPN axon collaterals in the GPe bear an axonal copy of motor signals sent to the SNr. This projection supports motor control by

inhibiting its own GPe circuit involving motor-suppressing Npas1 neurons in vivo. We propose a model by which dSPN GPe terminals act in concert with the canonical terminals in the SNr to control motor function via a striatopallidal Go pathway that involves at least to some extent Npas1 pallidostriatal projections. Specifically, we find that dSPN GPe collaterals track the speed of animals locomoting in the open field in a continuous manner. In a rotarod task, they also show sustained activation during the task sequence and acute activation at the temporal boundaries of individual motor bouts (up at onset, down at offset), consistent with previous findings at the cell body[40,43,48,49]. In our manipulation experiments we find that dSPN GPe collateral inhibition affects locomotor speed in the open field and motor performance in the rotarod. It would thus be interesting to determine if these behavioral phenotypes are related. For instance, it is possible that speed-related activity in dSPN GPe collaterals contribute to rotarod function. In order for animals to stay on the rotarod at increasing speeds, they need to perform environment-dependent posture adjustment, i.e., to adjust spatial and/or temporal features of their jumps (either increase jump amplitude, i.e., jump further or decrease jump duration/speed, i.e., jump faster) in order to follow the increasing speed of the rotarod. We find that the duration of individual jumps decreases with increased rotarod speed and correlates highly significantly with the duration of individual dSPN GPe collateral Ca2+ transients. On the other hand, the amplitude of jumps did not increase with rotarod speed and correlated to a lesser extent with the amplitude of dSPN GPe collateral Ca2+ transients (see Supplementary Fig. S5C–D). Thus, it is possible that speed-related dSPN GPe collateral activity allows environment-dependent posture adjustment by adjusting the temporal features of jumps (speed or duration) as the speed of the rotarod increases. Future work could address these questions by using motor assays requiring finer motor control. It would also be worth determining whether dSPN GPe axon collaterals control other types of motor behaviors and in particular whether the recently described functional segregation of striatal subregions[70] is maintained in the GPe via topographically organized projections of dSPN GPe collaterals[16].

These data converge onto an overarching model in which the direct pathway controls motor function via its simultaneous influence

on three brain regions: the classical targets of dSPNs (i.e., EP, SNr) and the GPe. Here our pathway-specific inhibition manipulations showed that dSPN GPe terminals are necessary for motor control in healthy mice. These results argue against the notion that the different dSPN outputs are redundant, rather suggesting they act in a complementary fashion, influencing motor output by leveraging their own distinct subcircuits[38]. Which subcircuit is engaged in the GPe? Together with recent studies using slice physiology, whisker stimulation or optogenetic manipulations under anesthesia[16,17,19], our findings indicate that dSPN GPe terminals control motor function by inhibiting GPe Npas1 neurons during behavior. Thus, via this monosynaptic connection, dSPNs are granted unique access to a neural population, Npas1 neurons, which largely project to the striatum and have potential broad impacts on striatal outflow[21-23,71,72] (see below for a more detailed discussion on Npas1 cells). Since Npas1 neurons target both dSPN and iSPN dendrites in the striatum, Npas1 neurons have been proposed to work as a gain control to filter weak synaptic inputs to the striatum[21]. Here we speculate that bridging collaterals could act as a fine-tuning knob for this gain control, inhibiting Npas1 neurons to facilitate cortical recruitment of SPNs during ongoing motion. Another possibility is that dSPN neurons do not send blanket inhibition onto all Npas1 neurons but rather would be topographically organized. For instance, dSPNs and Npas1 neurons could be organized into functional units encoding specific motor patterns, where dSPNs would only inhibit Npas1 neurons positioned into the active unit, thereby generating a disinhibitory feedback loop. This would complement a model proposed for SPNs and their local collaterals in the striatum, where competing units are inhibited[73]. Moreover, recent work showed that iSPN opto-stimulation disinhibits Npas1 neurons in vivo via a disynaptic circuit[15]. Thus, dSPN bridging collaterals could represent a mechanism to balance Npas1 output by competing with iSPN GPe inputs at a local level. Future work will be instrumental to address these possibilities.

We should address some of the limitations of our work including the possible implication of other subcircuits or other cell types in the GPe. Originally, GPe neurons have been divided into two populations, prototypical neurons and arkypallidal neurons which differ by their firing patterns, projections and molecular markers (reviewed in ref. 61). One proposed organization has been to oppose neurons expressing Nkx2.1 which label prototypical cells to neurons expressing FoxP2 which label arkypallidal cells, but newer molecular classifications have also been used[16,20,22,23,59-61]. To label arkypallidal cells, we here chose to focus on the Npas1 marker as Npas1+ neurons were shown to receive strong input from dSPNs[16], suppress locomotion[15,63], and as we show here recapitulate the effects of bridging collateral inhibition. A major advantage of this approach is that unlike FoxP2-cre mice, Npas1-cre mice only express cre in the GPe and not the striatum which allowed us to cross them with Drd1-cre mice for dSPN manipulations. However, while about 55-70% of Npas1 neurons co-express FoxP2+[20,59,60,62], and are considered arkypallidal neurons projecting to the striatum[59], 30-45% of Npas1 neurons are FoxP2-negative but Nkx2.1-positive and Lhx6-positive and project to reticular thalamus, cortex and substantia nigra (mostly compacta)[60,64,74,75]. Hence these extra-striatal Npas1 circuits could also contribute to the behavioral effects of bridging collaterals. Furthermore, the contribution of prototypical cells remains to be clarified: indeed, recent work showed that dSPN input to PV neurons (which label most prototypical cells[61]) is weak[16] and that PV neurons inhibit locomotion[63]. Thus if dSPN bridging collaterals would act via inhibiting PV neurons, PV neurons should decrease and not enhance locomotion. On the other hand, a recent paper showed that bridging collaterals target a small population of Lhx6-negative parafascicular thalamus-projecting PV cells, but not STN/SNr-projecting PV cells, which regulate reversal learning, but not locomotion. Moreover, a small population of Lhx6+ cells that are negative for PV and Npas1[60,64] have not been tested as to whether they

receive dSPN input and their role in motor control is unknown. Finally, we observed here that ChAT neurons, which are targeted by dSPNs[65], do not recapitulate the motor effects of bridging collaterals. It is possible, however, that ChAT neurons mediate other behavioral effects of dSPN collaterals not addressed here.

Of note, our optogenetic inhibition studies in anesthesized mice could suggest that dSPNs inhibit more than the 30% cells formed by Npas1 neurons (plus about 5% of ChAT neurons) in the GPe since depending on the analysis method we observed inhibition in 30-50% of units. However, inhibited units may also arise from polysynaptic rather than monosynaptic inhibition which we cannot distinguish here (since we measure the effect of inhibition of neurons that are active at 10-40 Hz frequency). Polysynaptic effects may explain why in our previous study[6] activation of iSPNs with the same ChR2 setup led to inhibition of 70% of units, even though iSPNs should primarily target Nkx2.1+ or PV+ cells which represent only 40-60% of the GPe population[20,59,61,64,74,76]. Last, there are Drd1/Drd2 co-expressing neurons in the DMS that project exclusively to the GPe[26]. These neurons should be targeted by our approach using Drd1-cre mice. However, they only represent a small % of SPNs, and do not promote locomotion like bridging collaterals do[26]. Thus, while our data are consistent with arkypallidal neurons mediating the effects of bridging collaterals on motor function future studies will have to address the involvement of other GPe subcircuits in this effect.

Because axon collaterals are abundant throughout the central and peripheral nervous systems[67,68] and profuse within the BG itself[8,11], it is intriguing to speculate on the potential advantages of axon collaterals as opposed to separate neural populations. One of the most studied functions of axon collaterals is their ability to send an efference copy of motor instructions to two brain regions simultaneously, one to instruct motor output and one to inform sensory brain regions of the sensory consequences of the action[77,78]. The same concept could apply here, whereby synchronous neural replicas of a motor command could help coordinate the activity of distributed brain regions to produce an effective motor output[38]. Here, we show that dSPN axons send correlated activity to the GPe and SNr, effectively modulating two of the critical nodes within the BG at the same time. This could have multiple consequences on downstream circuitry. For instance, in the olfactory piriform cortex, a sparse collateral excitatory network can act as an amplifier to boost the recruitment of output pyramidal neurons despite not all cells receiving an odor input[79]. Similarly, dSPN activated by a specific cortical motor program could facilitate recruitment of more SPNs (including those receiving weak cortical input) via disynaptic disinhibition (dSPN-Npas1-SPN) and in turn amplify striatal output for this same motor program. In the cerebellum, Purkinje cells harbor a vast inhibitory collateral network which provide inhibitory feedback and form feedforward loops, allowing to delineate the spatial borders of incoming signals[80]. Similarly, speculating that dSPN-Npas1-dSPN circuits are topographically organized, feedforward loops generated among these neurons could delineate the spatial boundaries of recruited striatal dSPNs. Determining the extent to which bridging collaterals act via a pallidostriatal circuit, and understanding the impact of these circuits on the spatial and temporal organization of striatal activity could be addressed in future work.

While we did not analyze the specific information encoded by dSPN GPe vs. SNr terminals, we found that the GPe/SNr axonal copy was not exact, and correlation coefficients between GPe and SNr presynaptic terminal activity were reduced by 20-30% during the running phase of a rotarod motor task. This modulation dependent on the task condition suggests the existence of local regulatory mechanisms specific to the GPe or SNr that may allow region-specific divergence of activity in a behaviorally relevant manner. This could arise from differential presynaptic regulation of terminal activity. For example, previous work showed that dopamine facilitates synaptic transmission at SNr, but not at GPe dSPN terminals via D1 receptors[81], possibly due

to terminal-specific trafficking of D1 receptors[36]. Moreover, synaptic transmission at dSPN terminals in the SNr is regulated by other mechanisms including GABA-B[27], cholinergic M4 receptors[82], CB1 receptors[83] and short-term facilitation[84]. Whether the same or distinct presynaptic mechanisms regulate dSPN GPe terminals remains to be determined. Moreover, the speed and amplitude of axon signals reaching GPe and SNr terminals could also be regulated by the molecular makeup of terminals or the biophysical properties of axons, e.g. myelin sheet properties, ion channel composition or geometry/length of collaterals[34,68,85]. For example, the arborization of dSPN collaterals in the GPe but not SNr is regulated by D2Rs[6], which could shape the propagation of electric signals. Determining how GPe vs. SNr dSPN terminals differ could also hold relevance for understanding ventral striatal circuits which, similar to the dorsal striatum, send axon collaterals to the VP on their way to the VTA[56,86]. Future work should also address whether other neuropeptides released by dSPNs (dynorphin[87], substance P[83,88]) are differentially released at GPe/VP and SNr/VTA terminals to regulate local circuits.

Bridging collaterals could represent interesting targets for therapeutic treatments. Indeed, promising new work suggests that dSPN inputs to the GPe could partly mediate the beneficial effects of GPe deep brain stimulation in a parkinsonian model[75]. Moreover, our previous work showed that regulating the density of bridging collaterals represents a longer-lasting mechanism to control the functional balance of the BG[6]. Indeed, the density of bridging collaterals is regulated bidirectionally by dopamine D2R levels, dopamine, SPN excitability or activity[6,7,16,24]. Hence, future studies could look into the molecular underpinnings of bridging collateral growth and retraction; how this impacts behavioral function, and how this applies to mouse models and disorders with altered motor function such as Parkinson's disease[16].

## Methods

### Reagents

References for all reagents (antibodies, AAVs, chemicals, mouse lines), data and equipment (hardware, software) is given in Supplementary Dataset S1.

### Mice

Animals were housed under a 12-h light/12-h dark cycle in a temperature-controlled environment (22 °C, humidity of 30–70%) with food and water available ad libitum, unless otherwise noted. Adult (>8 weeks old) male and female Drd1-cre (FK150Gsat/Mmcd; MMRRC), Npas1-cre-tdTomato (027718; Jackson; gift from S. Chan) and ChAT-cre (GM60; GENSAT) mice backcrossed onto C57BL/6J background were used for experiments. Findings for all figures apply to both sexes. Sex was not considered in study design nor is there enough animals per group to study interactions between sex and other variables. Sex-based analyses was not done because on original pilot experiments, no sex-differences were detected. Behavior testing was done in the light phase unless otherwise indicated. All animals for behavior were handled and habituated to tethering for 6–8 days. In each cohort, mice were used in several behavioral tests unless otherwise indicated. All animal procedures followed NIH guidelines and were approved by the New York State Psychiatric Institute or the National Institute on Drug Abuse and Johns Hopkins Medicine Animal Care and Use Committees.

### Surgical procedures

Stereotaxic coordinates, AAV volumes, AAV titers, and animal numbers (N) are given in Supplementary Table S1 and Supplementary Table S2. For retrograde tracing, Drd1-cre mice were injected with red retrobeads (1:3 diluted) into the SNr and a cre-dependent herpes-simplex virus (HSV)-GFP retrograde virus (1:2 diluted) into the GPe, and perfused 12 days later. For photometry, Drd1-cre mice were injected

unilaterally into the DMS with cre-dependent AAVs expressing jGCaMP7s[47] (1:4 diluted), Synaptophysin-GCaMP6s (SyGCaMP6s)[25] or Synaptophysin-jGCaMP8s (SyGCaMP8s). The SyGCaMP6s plasmid was obtained from O. Yizhar, cloned in-house, and sent for AAV production (Virovek). The SyGCaMP8s construct was designed in-house, then cloned and sent for AAV production at the ETH Viral Core (VVPP). Drd1-cre:Npas1-cre mice were injected with cre-dependent AAVs expressing ChrimsonR into the DMS and GCaMP6s into the GPe. In the same surgery, mice were implanted with GPe and/or SNr optic fibers fixed in place with superglue, dental cement and miniscrews. For chemogenetic/optogenetic manipulation, Drd1-cre mice were injected bilaterally with a cre-dependent AAV expressing hM4D[89] or mCherry into the DMS + DLS or with a cre-dependent AAV expressing eOPN3 (1:10 diluted) (gift from O. Yizhar;[58]) or YFP into the DMS. In the same surgery, mice were implanted with GPe fluid cannulas or optic fibers. Npas1-cre or ChAT-cre mice were injected bilaterally with a cre-dependent AAV expressing ChR2 or YFP and implanted with GPe optic fibers. For local [3H]-CNO infusion, anesthetized WT mice were implanted with the same guide cannulas used for behavior and [3H]-CNO infused during surgery the same way as for behavior (s. below). For in vivo physiology, Drd1-cre mice were injected with a mix of cre-dependent AAVs (1:1) expressing ChR2 and hM4D. Experiments began 4–6 weeks after surgery except for SyGCaMP8s (experiments done 10–14 days after surgery to ensure expression largely restricted to terminals, as in ref. 25). This study generated one original construct AAV9-CAG-DIO-Synaptophysin-jGCaMP8s (SyGCaMP8s), available from the lead contact upon request; the full sequence is provided in Supplementary Note 1. Validation of fiber and cannula locations is found in Supplementary Fig. S10.

### Neuroanatomy

Mice were transcardially perfused with ice-cold 4% paraformaldehyde in PBS under deep anesthesia. Brains were harvested, post-fixed in PFA overnight and washed in PBS. VGAT antigen retrieval was done by incubating brains overnight in 0.1 M Na-citrate buffer (pH 4.5), then heat-treatment for 60 s (600–800 W) in citrate buffer. Free-floating 30 to 50-µm coronal sections were cut using a Leica VT2000 vibratome. For staining, sections were incubated in blocking solution (5% fetal bovine serum, 0.5% bovine serum albumin in 0.5% PBS-Triton X-100) for 1 h at RT, and labeled overnight at 4 °C with primary antibodies against GFP (1:1000), DsRed (1:500), mCherry (1:500), Cre (1:1000) or VGAT (1:500). Sections were washed, incubated for 1 h at RT with fluorescent secondary antibodies (1:1000), mounted and coverslipped with Vectashield medium. Digital images were acquired using a Zeiss epifluorescence microscope or a Leica SP8 confocal microscope and processed with ImageJ. For Synaptophysin-GCaMP6s integrated optical density quantification, GFP fluorescence corresponds to immunostained GFP (GFP contained in GCaMP6s): total dSPN Synaptophysin-GFP+ terminal optical density in the region was quantified in ImageJ using two random counting frames per section positioned above the GPe or SNr (average 5.5 sections/brain region/animal); values reported are in percentage of striatal optical density as in ref. 6. Note that here GFP expression is targeted to the pre-synapse thanks to the Synaptophysin construct, whereas in Cazorla et al.[6], GFP was unspecifically expressed in dendrites/cell bodies in the striatum and axons in the projection fields. This explains the differences in GPe/SNr % of terminal field (arguably the present quantification is more accurate). For Synaptophysin-GCaMP8s vs. jGCaMP7s bouton vs. axon quantification, brains were kept unstained to compare native GFP fluorescence (GFP contained in GCAMP); optical density was calculated by selecting ROIs of boutons and axons (average 3.5 ROIs/section/brain region; 3.5 sections/animal); values reported are ratios of boutons vs. axons in same section. Note that TdTomato expression of the Npas1-2A-Tdtomato transgene is too low to be detected without antibody amplification.

## Fiber photometry during behavior

Fiber photometry equipment was set up using two 4-channel LED drivers connected to two sets of a 405 and a 465 nm LEDs (Doric). The 405 nm LEDs were passed through 405–410 nm bandpass filters, while the 465 nm LEDs were passed through a 460–490 nm GFP excitation filters using two 6-port Doric minicubes. The 405 and 465 LEDs were then coupled to a dichroic mirror to split excitation and emission lights. Low-autofluorescence patch cords (400 µm/0.48 NA) were attached to the cannulas on the mouse's head to collect fluorescence emissions. Signals were filtered through 500–540 nm GFP emission filters via the same minicubes coupled to photodetectors. Data were sampled at 1017.3 Hz. Signals were sinusoidally modulated, using Synapse software and RZ5P Multi I/O Processors (Tucker-Davis Technologies), at 210 and 330 Hz (405 and 465 nm, respectively) via a lock-in amplification detector, then demodulated and low-passed filtered at 3 Hz on-line (one-way 2nd order Butterworth filter). 405 and 465 nm power at the patch cord were set to 30 µW. For optogenetic stimulation, amber light (595 nm LED) was applied through the same optic fiber. The 595 nm light was passed through a 580–680 nm F2 port (photodetector removed) of the same 6-port minicube. Behavior was video-recorded with USB-cameras (Logitech) controlled by the Synapse software and frame timestamps recorded for post-hoc dataset alignments. Recordings were done as animals explored an open field arena (42 × 42 cm, Kinder Scientific) for 15 min or ran on a rotarod (UgoBasile). Rotarod testing consisted of trials at accelerating speed (5–40 rpm, until the animals fell, max 5-min) or constant speed (5, 10 and 15-rpm). Data analysis focused on the last 2 min of each trial when behavior was more stable (excluding the first/last 10 s in each epoch). Trials were separated by 10–12 min. Start and end of trials were timestamped in Synapse.

## Opto-photometry during behavior in closed loop

Animals were tested in the dark phase and mildly food-deprived to elicit locomotion. In a first experiment, dSPNs were stimulated in closed-loop based on Npas1 activity. Animals explored the open field for 1 h or when 12 trials per condition were completed. The opto-genetic LED was driven at various powers (0.5, 1, 2 mW), duration (3, 10 s) and frequencies (10, 20 Hz). At least 8 trials per condition were completed. Optogenetic stimulation was triggered when Npas1 calcium activity (dFF) reached a peak and was above a minimal threshold (estimated post-hoc to be around 2-zscores) for 250 ms. This was done using a custom-written Synapse program. dFF was estimated online using the following equation: $[(F-Fo) \div Fo]_{465} - [(F-Fo) \div Fo]_{405}$ where F is the fluorescent signal and Fo is the baseline 465 or 405 signal calculated using a sliding average window on the past 120 s. dFF was also low-passed filter at 1 Hz to avoid detecting irrelevant peaks. The computations took 200 ms on average to complete, hence the slight delay between the dFF peak and the beginning of the stimulation. In a second experiment, dSPNs were stimulated in closed-loop based on the animal's ongoing speed using a custom-written Anymaze protocol and the Anymaze AMi-2 optogenetic interface. Animals explored the open field for 1h20 or when 15 trials per condition were completed. Optogenetic stimulation (0.2 mW, 20 Hz, 5 s) was triggered only after a brief rest period (5 s) followed by a longer high mobility bout (average speed threshold reaches above 0.08 m/s and does not drop below 0.03 m/s for 5 s) were detected. At least 10 trials per condition were completed. In a third experiment, we verified the absence of crosstalk between the 465 and 595 nm LEDs. First, we checked that the Npas1 GCaMP photometry signal in mCherry controls was unaffected by optogenetic 595 nm light at all of the powers tested (2 mW or lower). This confirmed that the 595 nm LED does not significantly activate GCaMP under these conditions. Second, we verified that the 465 nm photometry LED does not promote dSPN-induced behavioral changes, using rotations as a readout. We activated the 465 nm LED for

10 s, 20 Hz (4 trials each, 40 s ITI) at 0.03 (photometry power) or 0.35 mW (>10x higher than photometry power).

## Local chemogenetics during behavior

Saline or CNO (300 nL, 1 mM as in ref. 90) was locally infused in the GPe at an average rate of 0.1 µl/min through internal cannulas connected to 2 µL Hamilton syringes via PE50 tubing calibrated for volume. Correct infusion was verified post-infusion. Infusion cannulas were left in place for 18 min then removed. Behavior started 5 min after. Experiments were separated by minimum 2 days to avoid carry-over effects of CNO. Because cannulas quickly get clogged after a few infusions, we prepared several cohorts. Some mice were tested only in open field, some only in rotarod and some in both. Since CNO has a long half-life (several hours[91]), saline/CNO comparisons cannot be performed on the same day (like we did for optogenetic experiments). For rotarod, mice were pre-trained to reach stable performance (3-4 trials/day for 6 days), then tested in constant speed trials (40 rpm) after Saline or CNO infusion. Latency to fall and total number of falls were recorded. For open field, mice were tested after Saline or CNO infusion on separate days. Locomotion was measured using infrared beams (Motor Monitor, Kinder Scientific).

## Radioactive CNO infusion and autoradiography

Radioactive [3H]-CNO (70 Ci/mmol) (1 mCi/mL; 14 µM) was obtained from Novandi (Sweden) in pure 99% EtOH. [3H]-CNO was diluted 1:140 to a final concentration of 7 µCi/mL; 0.7% EtOH; 100 nM and locally infused in the GPe of anesthetized WT mice through cannulas at an average rate of 0.1 µl/min. The cannula was left in place for 15 min. It was not possible to use the same CNO concentration as for behavior because the solution needed to be diluted to acceptable radioactivity and EtOH levels, but the same volume (300 nL) was used. 30 min after infusion (=time of behavioral testing), brains were collected, flash frozen and stored at −80 °C. Tissue was sectioned (20 µm) on a cryostat and thaw mounted onto ethanol-washed slides. Slides were air dried overnight, placed in a Hypercassette™ and covered with a BAS-TR2025 Storage Phosphor Screen. Slides were exposed to the screen for 12–14 days and imaged using a phosphorimager (Typhoon FLA 7000).

## In vivo electrophysiology

Anesthetized mice (chloral hydrate) were locally infused with Saline or CNO (300 nL, 1 mM) at an average rate of 0.1 µl/min. Mice were then implanted with an optic fiber into the site of ChR2 injection in the DMS. A glass electrode (impedance 8–12 MΩ) filled with 2 M NaCl was lowered into the GPe or SNr. The electrode was lowered using a manual hydraulic micropositioner to detect spontaneously active neurons. Recordings started at the minimum 25 min post-Saline or CNO infusion. From this starting point, the GPe or SNr was sampled in four locations spaced 0.15 mm apart and arranged in a 2 × 2 spaced grid moving in a clockwise direction. The starting locations were counterbalanced across animals and groups. GPe and SNr neurons were identified using a combination of stereotactic position and narrow action potential width (<1 ms). After 2–3 min of stable recording, optical stimulation (473 nm; 2 mW) was applied for 0, 250, 500 or 1000 ms (10 sweeps at each stimulation) in a pseudo-randomized order as recording continued. Neuronal activity was amplified and filtered (1000x gain, 100–10 K Hz band pass) and fed to an audio monitor and to a computer interface with custom-designed acquisition and analysis software (Neuroscope). Traces from continuous recordings were analyzed offline, first by applying a window discriminator to identify spikes, then from the spike table to construct Peri-stimulus time histograms (PSTHs). PSTHs were constructed by sampling spikes with 1 ms bins, 1000 ms before and 2000 ms after laser illumination, and by summing data from 10 sweeps (shown as spike frequency (Hz)). Neurons that decreased

or increased their response by 1/3 of the pre-stimulation firing activity in the full laser stimulation period (250, 500 or 1000 ms) or in the first 50 ms (separate analysis show in in Fig. 6J, L) were labeled as being significantly inhibited or excited, respectively. Normalized spike frequencies were calculated by dividing the mean activity during the stimulation epoch [250, 500 or 1000 ms (& 1000 ms for 0 ms condition)] by the mean of baseline activity obtained during the 1000 ms preceding laser illumination. They were analyzed for all units and for the significantly excited units, separately. In a separate analysis, Z-score of spike frequency was also calculated as: $[(mean_{stim\ period}) - (mean_{baseline\ period})] \div (SD_{baseline\ period})$. A Z-score of −2 and +2 was used as the cut-off value to define cells that are significantly inhibited or excited, respectively. Basal spike frequency was calculated by averaging the spike frequency in the 1000 ms preceding laser stimulation. An average of 50 neurons were recorded per condition (8–10 neurons per mouse, 5 mice per condition; GPe and SNr recorded in separate mice).

## Optogenetic manipulations during behavior

Self-made (ThorLabs and Precision Fiber Products) or commerical (Newdoon) optic fibers with minimal average 80% transmittance were used. Optical stimulation was provided by a laser emitting light (473 nm for ChR2, 532 nm for eOPN3) activated with an Arduino microcontroller. Lasers were connected via optical patch cords (200 µm, 0.22 NA) and a rotary joint to the animals' optic fibers. To activate ChR2, we used 10 ms light pulses at 10- or 20-Hz (6–10 mW at fiber tip). To activate eOPN3, we used 200-ms light pulses at 1-Hz (20% duty cycle) (4–6 mW at fiber tip). Since some cement caps fell during experimenting, some mice were tested only in open field, some only in rotarod and some in both. For rotarod, animals were briefly habituated to rotarod running for 3 days to minimize stress. Testing consisted of 6 trials (5–40-rpm accelerating speed, max 10-min) separated by 10–12 min with the following schedule: 2 laser-off, 2 laser-on, 2 laser-off trials. Latency to fall was recorded. For open field in open-loop (ChR2), 10 laser-on trials of 10 s each were delivered at 40 s intertrial interval as in ref. 63. Total locomotion was measured using infrared beams. For open field in closed-loop (eOPN3; testing done in dark phase), 4–5 laser-on trials and 4–5 laser-off trials of 30 s each were delivered at 6 min intertrial interval (maximal test duration 90 min). The long intertrial interval was necessary to allow eOPN3 to recover as shown in ref. 58, which we also observed here (Fig. 7D). Stimulation was done in a closed-loop fashion using a custom-written Anymaze protocol and the Anymaze AMi-2 optogenetic interface. The trial started only after a brief rest period (2 s) followed by a longer high mobility bout (speed threshold reaches above average 0.08 m/s and does not drop below 0.03 m/s for 5 s). See increase in locomotion in GFP and eOPN3 mice in the pre-stimulation epoch (Fig. 7D).

## Machine learning-based videography

DeepLabCut[50] (DLC) was used for tracking mouse body parts during behavior. DeepLabCut 2.1.8.2 (computer) and 2.1.10.2 (googlecolab) were used using default parameters and the pretrained resnet50 network with imgaug augmentation. Frames were extracted with the k-means method and outlier frames with the jump method. *Open field:* 8 body parts (snout, both ears, body center, both side laterals, tail base and tail end) and the 4 openfield corners were manually labeled. An initial 380 frames (20 images from 19 videos) were used to train a network for 170 K iterations. 20 outlier frames and 380 new frames were (re)labeled to refine the network for 210 K iterations (from scratch). 300 new frames were labeled to improve the pixel error to a final 400 K iterations (train error: 2.65, test error: 3.71). *Rotarod:* 5 body parts (2 paws, 2 ankles, tail base), 4 corners of the rotarod, 2 points on the rotarod wheels and 4 points in a flashing LED (timestamps) were manually labeled. An initial 180 frames (20 images from 9 videos) were used to train a network for 80 K iterations. 280 new frames were

labeled to refine the network for 200 K iterations (from scratch) (train error: 3.00, test error: 3.75). DLC data was aligned with other data using videoframe timestamps provided by TDT Synapse, Anymaze or Arduino. Pixels were converted to cm using known distances: open field corner (42 cm) or rotarod height (3 cm). DLC data was upsampled to 100 Hz to align with the calcium data (inpaint_nans function in Matlab; John D'Errico) and further processed as described below. DLC networks are available (see section Data availability below). Representative videos (Supplementary Videos) were generated using custom-written Python scripts and Adobe Creative Cloud Express.

## Fiber photometry data analysis

Data was analyzed using custom in-house Matlab scripts. *Preprocessing:* Data was 10x downsampled to 101.73 Hz. The first 3–5 min were trimmed. Change in fluorescence, dFF (%), was defined as $100 \times (F - F0) \div F0$, where F represents the fluorescent signal (465 nm) at each time point. F0 was calculated by applying a least-squares linear fit (polyfit) to the 405 nm signal to align with the 465 nm signal. For rotarod data, F0 was computed on the baseline pre/post period only. To normalize signals across animals and sessions, we calculated modified z-scores using the median absolute deviation (MAD): $zscore = 0.6745 \times (dFF - median_{dFF}) \div MAD_{dFF}$. For the rotarod, data was baseline corrected by substracting the 8th percentile of the data calculated in a moving window across the baseline pre/post sections (to not deform the data). For peak frequency and amplitude and interpeak interval calculation presented in Supplementary Fig. S4D–E, dFF was low-pass filtered using a 0.5 s moving average smoothing to avoid overcounting peaks not relevant to the behavior of interest (jumps). We verified the filter did not alter the data (Supplementary Fig. S11). *Open field with jGCaMP7s:* Mouse body center speed was smoothed with a 2 s window. Points with likelihood <0.9 were interpolated with inpaint_nans function. Locomoting bouts were identified as epochs when mouse speed broke 5 cm/s. Movement onset was identified by looking back to find the local minima (findpeaks function) just prior to when speed broke 3 cm/s. Movement offset was first identified as timepoints after speed went back below 3 cm/s. The exact end was identified by looking back to find the local maxima just prior to when speed broke 5 cm/s on the descending phase. Calcium data were aligned to movement onset/offset. Local minima/maxima were computed in a 0-to-5 s timewindow after onset/offset. Data amplitude change was computed by substracting to each other the relevant local minima and maxima values closest to the beginning and end of the respective motor bouts. Pearson r correlation between calcium data and smoothed mouse speed (or between GPe and SNr data together) were calculated across the entire dataset. We used a lag analysis by temporally shifting one dataset behind the other to identify maximal points of correlation. We report the strongest correlation. Statistics were computed by comparing the populations of maximal correlation value of the real datasets vs. 1000x shuffled datasets where one of the variables to correlate was phase-shuffled (PhaseShuffle function in Matlab; Edden M. Gerber). *Open field with opto-photometry:* Calcium data was baseline corrected against the dFF average in the −15 to −5 s timewindow before opto-stimulation onset and aligned to opto-stimulation onset. Amplitude change in dFF was computed by substracting value at opto onset to value at opto offset. Average speed was calculated in the opto-epoch, in the baseline epoch (−15 to −5 s) and in the speed thresholding epoch (−5 to 0 s) in Fig. 8I, J. Note: during data processing, we verified that z-scoring did not qualitatively affect the final results. To quantify body rotations, normalized body vectors were generated from lower body and snout positions. Body vectors at consecutive timepoints were designated as v1 and v2, with vector coordinates (x1,y1) and (x2,y2), respectively. Rotation angle in this period was calculated as follows: $angle = \arctan 2((x1 \times y2) - (y1 \times x2), (x1 \times x2) + (y1 \times y2))$. The angle

was converted from radian to degree (°). Cumulative degrees rotated were calculated from opto-stimulation onset. *Rotarod with jGCaMP7s:* Basic analyses were done on pre, during and post epochs and included: baseline levels (8th percentile calculated in a moving window), peak frequency (findpeaks function), peak amplitude (compared to local minima), AUC (trapz function; normalized to time) and interpeak interval (in s). To characterize mouse behavior during rotarod running, we chose to use the Y position of the lower body because (1) the lower body had excellent tracking dynamics, which was not the case for the feet (Supplementary Fig. S5A) and (2) the lower body movement dynamics in relation to rotarod behavior were best explained by position changes in the *Y* axis (jump up and down the rotarod), rather than variations in the *X* axis. The Y position (Ypos) of the lower body was z-scored using the following formula: $zscore_{Ypos} = 0.6745 \times (Ypos - median_{Ypos}) \div MAD_{Ypos}$. Jump onset was calculated using the findpeaks function on the inverted z-score$_{Ypos}$ to find position minima, i.e., when the animals were at the lowest point on the rotarod and about to jump. Similarly, jump offset was calculated using the z-score$_{Ypos}$ to find position maxima, i.e., when the animals were at the highest point on the rotarod and about to slide back down on the rotarod. Data with likelihood <0.95 (DeepLabCut metric) was excluded from analysis. Epochs of at least 1 s with repeated points with likelihood <0.95 were also excluded. Sessions with <50% points with likelihood >0.95 were excluded. GPe/SNr calcium data were aligned to the onset/offset of lower body jumps. Calcium events were normalized to the data in the −5 to −1 s period prior to jump onset. Event duration was computed from event start to peak maxima (for movement onsets) or peak minima (for movement offsets) detected by looking for the closest local maxima/minima to the event in the 0–5 s timewindow. Pearson correlations were computed between individual body parameters (jump duration, from peak to the next peak) and dFF parameters (dFF transient duration, from peak to the next peak) using all trials for all mice pooled together. Finally, Pearson r correlation coefficients with lag analysis (see above) between GPe and SNr calcium data were calculated across entire epochs (pre, during or post rotarod). Figures were generated by aligning datasets to rotarod on and rotarod off (we included NaNs in the middle during concatenation).

### Closed-loop optogenetic assay and behavioral classification

Relative body center mouse speed was calculated by substracting the baseline speed calculated in the pre-stimulation epoch (−60 to −5 s). DeepLabCut body points with likelihood <0.9 were interpolated with inpaint_nans function in Matlab. Locomotion frames were defined as frames when body speed reached >4.5 cm/s. Motionless frames were defined as frames when the speed of all body parts was ≤0.8 cm/s. Other non-locomotor frames were defined as frames not falling into the other categories. Frame category were used to create behavioral maps in the pre-stimulation and during-stimulation periods and percent time in each category quantified. The arena was subdivided into a periphery zone (most outer 10 cm square) vs. center to classify frames (periphery or center) as a proxy for anxiety.

### Statistics

Full statistical results including *p*-values and *F*-values can be found in Supplementary Dataset S2. Statistical analyses were performed using GraphPad Prism 8 or MATLAB. Data are expressed as mean ± standard error of the mean (SEM). Pearson *r* correlation coefficient populations obtained from real vs. shuffled data were compared with a non-parametric Mann–Whitney test. Paired and unpaired two-sided Student's *t*-tests were used to compare all other two-group data. For in vivo physiology data, distribution of significantly inhibited and non-inhibited units was compared using Fisher's exact test. Multiple comparisons were evaluated by ANOVA and Tukey (one-way ANOVA) or Sidak (two- or

three-way ANOVA) post hoc tests using correction for multiple comparisons; only when interactions were significant. Mixed ANOVA models replaced two- or three-way ANOVA if dataset was not fully balanced. A *p*-value of <0.05 was considered statistically significant.

### Reporting summary

Further information on research design is available in the Nature Portfolio Reporting Summary linked to this article.

### Data availability

Source data are provided with this paper. DeepLabCut datasets are deposited at Zenodo under accession code [https://doi.org/10.5281/zenodo.6448813][92] and [https://doi.org/10.5281/zenodo.6448595][93]. Please cite the code's DOI and this paper upon usage of datasets. Any additional information or raw datasets required to reanalyze the data reported in this paper is available from the lead contact upon request. Source data are provided with this paper.

### Code availability

Original codes are deposited at Figshare under accession code [https://doi.org/10.6084/m9.figshare.23609595.v5][94]. Please cite the code's DOI and this paper upon usage of the code.

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

## Acknowledgements

The study was funded by the NIMH (R01 MH093672, R01 MH124858 to C.K.), the Swiss National Science Foundation (SNSF) (P2EZP3_168841, P400PB_180841 to M.A.L.), NINDS (R01/R56 NS069777 to C.S.C.), NIDA (ZIA000069 to MM) and the Philippe Foundation (awarded in 2017, 2018 to M.A.L.). M.A.L. is supported by an Ambizione Grant from the SNSF (PZ00P3_193430), a NARSAD Young Investigator Grant from the Brain and Behavior Research Foundation (30854), a Young Investigator Grant from the Novartis Foundation for Medical-Biological Research (22B097), and Research Grants from the Vontobel Stiftung (1334_2021), the Olga-Mayenfisch Stiftung (awarded in 2022) and the Neuroscience Center Zurich (awarded in 2021). A.T.H. is supported by the European Molecular Biology Organization (ALTF 561-2020), J.V. by the Leon Levy Foundation (awarded in 2017) and the NIMH (T32MH018870) and J.B. by the MICIN (RYC2019-027371-I and PID2020-117989RA-I00). O.Y. is supported by the European Research Council (819496), the EU Horizon2020 program (H2020-ICT-2018-20 DEEPER 101016787) and by the Israel Science Foundation (3131/20). We thank C. Labouesse, T. Rahbek-Clemmensen, S. Gershbaum, G. Stevens, B. Rao and M. Billiard for advice with analysis or scripts, A. Cebula, J. Sherman and J. Baer for technical assistance and S. Modica (ETHZ VVPP) and M. Mahn for help with construct design/

cloning and AAV production. Figures 1A, E; 2A, C; 3A, 4A, D; 5A, D, E; 6A, 7A, C, D; 8A, C, H; 9A, C, D, E; S3A; S4A; S5A; S9B were created with BioRender.com.

## Author contributions

The study was conceived by C.K. and M.A.L. M.A.L. did surgeries, in vivo recordings, chemogenetics, optogenetics, behavior, neuroanatomy and designed the SyGCAMP8s construct. M.A.L., A.T.H., and J.G. wrote Matlab or Python scripts and analyzed data. M.O.C. did in vivo physiology including surgery and analyses, which C.K. and J.V.-V. supervised. M.A.L., X.S., A.T., and J.G. made DeepLabCut networks. M.A.L. and C.L. built in vivo behavior rigs. J.V. and M.A.L. did confocal imaging. J.V., M.Z., and J.G. helped with anatomy or behavior. S.L. and J.B. did autoradiography, which M.M. supervised. S.C. provided Npas1-cre mice and key scientific input. O.Y. provided SyGCaMP6s and eOPN3 constructs and key scientific input. C.K. supervised the study. C.K. and M.A.L. acquired funding. M.A.L. made figures. M.A.L. and C.K. wrote the paper with inputs from all authors.

## Competing interests

J.B. and M.M. (individuals) are listed as inventors on an application (62/627,527) filed with the U.S. Patent Office regarding DREADD compounds, but none of these compounds are used in the manuscript. M.M. has received research funding from AstraZeneca, Redpin Therapeutics, and Attune Neurosciences. J.V.-V. has served on advisory boards for Roche, Novartis, and SynapDx; has received research funding from Roche, Novartis, SynapDx, Forest, Janssen, Yamo, MapLight, and Seaside Therapeutics; and has received editorial stipends from Wiley and Springer. O.Y. (indiviual) is listed as an inventor on a patent application (US20210403518A1) filed with the US Patent Office regarding the optogenetic tool eOPN3 used in Fig. 7 and Fig. S8, and serves as a consultant for Modulight.bio. Remaining authors declare no competing interests.
