## [Peer Review File · Nature Communications]

A non-canonical striatopallidal Go pathway that supports motor controlREVIEWER COMMENTS

Reviewer #1 (Remarks to the Author):

The paper submitted by Labouesse et al., investigates the functional impact of the bridging collaterals formed by direct striatal projection neurons (dSPNs) into the GPe. The authors show that these collaterals send a copy of the motor signals, classically send to the SNr, encoding motor parameters such as the body speed in an open field or temporal aspects of motor bouts when mice are running on a rotarod. The authors described this pathway as being a non-canonical 'Go' pathway necessary for motor function. The proposed circuit mechanism in the GPe goes through the inhibition of Npas1 pallidostratial 'stop' neurons that would result in a positive gain for striatal activity. Strengths of this paper include the combined anatomical and thorough functional characterization (assessed using global calcium fiber photometric signals) of dSPNs terminals in the GPe vs. the ones in SNr. In particular, the approach to dissect the functional contribution of the GPe bridging collaterals using chemogenetic and closed-loop optogenetic stimulation is elegant and well executed. In addition, the in-vivo electrophysiological approach to test the specificity of the chemogenetic manipulation of dSPNs GPe terminals is convincing and a nice validation of the approach. The weakest point of this work is the proposed circuit mechanism acting through 'pallidostratial' Npas1 neurons. Indeed, the evidence to solely rely on a pallidostratial circuit mechanism is rather weak. Altogether, I still believe that the paper is important, timely, and of broad interest to the field.

I found this paper very well written yet, in this work, the authors associate Npas1+ GPe neurons with FoxP2+ arkypallidal neurons. For example, the authors state that 'Npas1 in the GPe is almost exclusively expressed in arkypallidal neurons'. This concept is not accurate and sends a confusing message to the field. Indeed, arkypallidal FoxP2+ neurons only represent a subclass of Npas1+ cells. In fact, as already described in multiple studies, many Npas1+ neurons are not arkypallidal neurons with proportions ranging from 40% (see Dodson et al., PMID: 25843402; Cui et al., PMID:33731450) to 50% (see Abrahao et al., PMID: 29917235). This discrepancy between Npas1+ and arkypallidal neurons should be clearly stated and the manuscript revised accordingly.

My previous comment lead to a second consideration regarding the circuit mechanism that might underlie the functional effect observed after inhibition of the dSPNs bridging collaterals in the GPe. Currently, the authors interpret their results through the contribution of pallidostratial circuits. This is partly justified by the fact that all Npas1+ neurons display pallidostratial projections. However, as mentioned above, a substantial proportion of these neurons are non-arkypallidal Npas1+ neurons that also send projections to classic targets of GPe neurons (such as the STN and the SNr), but also to unconventional targets (such as the cortex, see Abecassis et al., PMID: 31811030). The functional impact and contribution of all these different projections formed by Npas1+ neurons are not considered in this work. This represents an important limitation that should at least be acknowledged and thoroughly discussed in the manuscript.

The authors mentioned that dSPN inputs inhibit Npas1+ or FoxP2+ arkypallidal neurons and cite three references to support this claim (Cui et al., 2021; Ketzef & Silberberg 2021; Johansson & Ketzef, 2023). However, work by Spix et al., (PMID: 34618556) has also described that these dSPN projections are not exclusive to arkypallidal neurons but also drive 'robust inhibition in Lhx6-GPe neurons' (which are STN/SNr downstream projecting neurons). Once again, this is problematic as it would suggest that the effect of dSPNs collaterals in the GPe could also be relayed by other means than pure pallidostratial circuits. It would be nice to discard these potential 'non-pallidostratial' effects (or at least mention their existence as potential drawbacks).

In agreement with my previous comment, figure 6G shows that a high proportion of GPe neurons (close to 50% of the recorded cells in the saline condition) are inhibited by dSPN optogenetic stimulation. This goes against dSPNs collaterals only impacting arkypallidal neurons (that represent less than 30% of all GPe neurons). Also, here the authors only make the distinction between 'inhibited vs. non-inhibited cells'. What do 'non-inhibited cells' include as neuronal

responses? For example, did the authors record cells that were excited by dSPN optogenetic stimulation?

As stated before, one limitation of this work is the lack of a precise circuit mechanism to explain the behavioral effect caused by dSPNs bridging collaterals inhibition in the GPe. In particular, the absence of supportive evidence in favor of a pallidostriatal circuit weakens the authors' conclusion. If time allowed, I believe that this could easily be addressed by a straightforward experiment showing that the striatal activity (measured through GCaMP signal in dSPNs) was truly altered during motor behavior following the inhibition of dSPNs bridging collaterals in the GPe.

The open field motor classification performed in Fig.7 E, F includes locomotion frames, motionless frames, and a category defined as 'fine movements' for video frames that did not fall into the 2 first categories. However, looking at the videos, I really wonder if the video's quality (and field of view) was sufficient to provide any information on fine movements. In addition, does this 'fine movements' category contain heterogeneous motor behaviors? If yes, wouldn't it fall better as a 'background/noise' category?

In the discussion p18/29, the sentence line 16-17: 'Moreover, recent work showed that iSPN opto-stimulation disinhibits Npas1 neurons in vivo via a disynaptic circuit' is citing the ref: Cui et al., 2021. However, please note that the cited paper does not contain in vivo recordings, nor does it reveal a disynaptic disinhibition of Npas1+ neurons. In fact, this work clearly shows (see Fig. 7d) and states the opposite: 'On the other hand, although there was a big difference in the strength of the DLS-iSPN input between PV+ neurons and Npas1+ neurons, we did not observe a difference ($p=0.083$) in the fold change of firing between PV+ neurons and Npas1+ neurons from DLS-iSPN stimulation...'. Please amend the text accordingly.

The authors reported the dilution of all the viral solutions used in this manuscript but, for best reproducibility, the stock titer should also be provided.

I could not find the stereotaxic coordinates used by the authors for all their experiments. I might have missed it but, if not, it would be nice to provide them directly in the methods.

Reviewer #2 (Remarks to the Author):

Here, Labouesse and colleagues performed a variety of experiments to test the functionality of "bridging" collaterals from direct pathway striatal projection neurons (SPNs) to the GPe in awake behaving mice. Specifically, a series of photometry, chemogenetic, and optogenetic experiments were carried out. These experiments potentially support two important phenomena: (1) the dSPN->GPe projection is relevant for motor behavior and (2) the effect of this projection is primarily mediated by Npas1 GPe neurons. The number of experiments and amount of work here is very impressive; however, I found the presentation confusing to follow, lacking some key details, and I did not fully understand a few essential controls. I am hoping that my confusion can be resolved through careful reanalysis of the data rather than carrying out any new experiments. If this can be resolved, then I see path for this manuscript being suitable for publication in Nature Communications.

Major points that must be addressed:

Use of the word collateral

Can the authors clarify a few basic points re: bridging collaterals? After reading through this manuscript and cited works I'm confused about whether *all* dSPN->GPe projections are in fact axon collaterals.

First a minor point: in Figure 1d, assuming I understand it correctly, these numbers are substantially larger than the prior report in Cazorla et al. (Fig. 1e). Can the authors comment on this?

Secondly, and more importantly, is there evidence of dSPNGPe projections that are not axon collaterals? Some of the cells in Figure 1E look like they express HSV and are not labeled by retrobeads – admittedly the picture is fuzzy in the PDF so maybe I'm missing something here. There is no quantification so I'm left wondering if there is a non-trivial number of dSPNs that project to GPe and not SNr. Also, if indeed some cells appear to only project to GPe, an alternative explanation is that some iSPNs are labeled due to viral expression leak – I do not know what the likelihood of this is with the HSV used by the authors. It's doubly hard to assess the likelihood of the latter possibility since the viral titers are not listed in the methods section (note that the same concern is there for AAVs due to not listing titers). I'm hoping this is a misunderstanding on my part and new experiments are not needed. Either way, this needs to be shored up since the author's interpretations rest on the idea that all dSPNGPe terminals come from axon collaterals.

Viruses used and signal conditioning

The authors convincingly show that terminal calcium signals in GPe and SNr are correlated – of course this should be the case if both originate from the same dSPNs. However, I'm concerned about a few things here that might be over-stating the result: (1) there is not a direct comparison between the synjGCaMP8s results and the jCaMP7s results on the rotarod (2) I could not find a characterization of synjGCaMP8s kinetics in the manuscript, and (3) the signals all arise from slow GCaMP variants and signals are heavily smoothed after digitization.

First, as the authors openly admit, Figures 2 and 3 arise from non-terminal targeted jGCaMP7s. This motivates them to develop a synjGCaMP based on jGCaMP8s, which is used in Figure 4 for a second stab on rotarod recordings. Rather than showing results in a separate figure, it would be more useful to directly quantify how much of the Figure 3 results are due to cross-talk from dSPN \diamond SNr axons. By eye it looks like there's a 15% reduction in Pearson correlation when targeting to terminals (i.e. it's a significant, but not huge contribution, which is reasonable). It is important to note that synjGCaMP targeting is likely not perfect (as the author show), and there is still probably a small contribution from dSPNSNr axons.

Second, there is no characterization of synjGCaMP8s kinetics. A good first step here is to report the autocorrelation in baseline data from all GCaMP variants (and targets where it was expressed) used in the manuscript. If synjGCaMP8s has substantially slower kinetics, then it's possible that synjGCaMP8s is under-reporting the difference between GPe and SNr terminal photometry signals.

Third, I'm hoping the authors can re-analyze a subset of the data with much less smoothing (see a related minor point below). The slow variants of GCaMP are already smoothing calcium activity (note: repeating these experiments with different GCaMPs is overkill in my opinion), so it is important to know whether the post-hoc smoothing is masking fast time-scale differences.

Controls for chemogenetic manipulations

I am confused about the controls for a key experiment – chemogenetic inhibition at dSPN axon terminals in GPe. Ideally CNO does not directly impact the dSPNSNr projection. It is unclear if this is just an unrepresentative example, but the baseline firing rate of the neuron shown in Fig. 6F (CNO) appears substantially lower than the neuron in 6E (SAL). Additionally the opto-response looks fairly blunted, especially in the early phase in response to the 250 ms laser pulse.

I recommend a careful reanalysis of this dataset. Is there a systematic shift in the baseline firing rate? I am also wondering if the normalization shown on the right side of Fig. 6 masks the blunting of the opto-response. How do these results look when analyzing the raw difference in firing rate due to opto-stim? I could not find details on calculation the normalized response, but what if the

normalized response is measured by taking $\min(\text{firing_rate})$ during stim and comparing to baseline? I wonder if some of the subtle effects are washed out by summing or averaging.

Npas1-cre/Drd1-cre experiments

I am hoping the authors can clarify this experimental configuration. First, two BAC transgenic mouse lines were crossed. Did the authors characterize this cross or has it been characterized in a previous publication? While the location of the Drd1-cre transgene is known, it looks like the location of Npas1-cre is unknown, so it is important to confirm that there is not an unwanted phenotype in the double mutant. Also the Npas1-cre line from MMRRC is listed as also containing tdTomato. Do the Npas1-cre mice also express tdTomato or am I misunderstanding something?

Second, are the authors certain that GCaMP did not express in dSTr axon terminals? The red channel is really saturated in the histology so I can't tell if any GCaMP virus was taken up by axons and (possibly) retrogradely transported. The reason I ask is that Figure 8D shows an apparent increase right after stim. It's possible this example is unrepresentative, but the increase seems to scale with power (especially in the first 10-30 trials). The authors should carefully look at this data, because a fast-timescale increase would suggest that photometry is both reporting an expected inhibitory response from Drd1GPe GABA release and the excitatory response in Drd1 axon terminals. If the authors performed this control already it would be useful to know what if any GCaMP signal is present in Drd1-cre only animals. If this data is not on hand, a careful re-analysis of their data is hopefully sufficient to resolve this. A good first-step here would be to look at un-zscored dF/F_0 for the same animal, take the difference between .5/2mW and 0mW and then average.

ChAT-cre experiments

In Figure 9 why was ChAT used a comparison and not PV? The expression level in ChAT neurons looks extremely low, so it's hard to tell if the difference in effect is due to cell-type targeting or simply number of cells expressing Chr2.

Minor points

- 1) Photometry experiments are referred to as "imaging". Since photometry is not image forming so I would replace "calcium imaging" with "calcium photometry".
- 2) Virus titers are not reported. This is important to know to enhance reproducibility and is an important detail since the interpretation of the results depends on ruling out any "leaky" expression of DIO constructs.
- 3) Checking to see if this is a typo, in Fiber photometry during behavior in methods the authors mention demodulating offline then applying a 3 Hz low pass filter and sampling the 3 Hz low-passed signal at 1017.3 Hz. Do I have that right that the signals were this over-sampled?
- 4) Related to the previous point, another smoothing filter is applied to the photometry data, either 1 H low-pass z or a .5s moving average. This sounds like the photometry signals are really smoothed. I'm surprised that the 3 Hz low-pass was not enough to remove high frequency noise. Can authors comment on this?
- 5) For all filters used in data pre-processing please specify the filter that was used and how it was applied to the data (e.g. if it was applied forward and backwards to remove phase distortions).

We would like to thank the reviewers for their thorough review of our paper “A non-canonical striatopallidal “Go” pathway that supports motor control” reviewed for *Nature Communications*, **Manuscript ID #NCOMMS-23-04116**. We were pleased to find that the reviewers found our study to be important, timely, impressive and of broad interest to the field. Please find below our point-to-point responses to the comments. Edits to the manuscript text can be found in blue.

Reviewer #1:

The paper submitted by Labouesse et al., investigates the functional impact of the bridging collaterals formed by direct striatal projection neurons (dSPNs) into the GPe. The authors show that these collaterals send a copy of the motor signals, classically sent to the SNr, encoding motor parameters such as the body speed in an open field or temporal aspects of motor bouts when mice are running on a rotarod. The authors described this pathway as being a non-canonical ‘Go’ pathway necessary for motor function. The proposed circuit mechanism in the GPe goes through the inhibition of Npas1 pallidostriatal ‘stop’ neurons that would result in a positive gain for striatal activity. Strengths of this paper include the combined anatomical and thorough functional characterization (assessed using global calcium fiber photometric signals) of dSPNs terminals in the GPe vs. the ones in SNr. In particular, the approach to dissect the functional contribution of the GPe bridging collaterals using chemogenetic and closed-loop optogenetic stimulation is elegant and well executed. In addition, the in-vivo electrophysiological approach to test the specificity of the chemogenetic manipulation of dSPNs GPe terminals is convincing and a nice validation of the approach. The weakest point of this work is the proposed circuit mechanism acting through ‘pallidostriatal’ Npas1 neurons. Indeed, the evidence to solely rely on a pallidostriatal circuit mechanism is rather weak. Altogether, I still believe that the paper is important, timely, and of broad interest to the field.

We thank the reviewer for this positive assessment. The focus of the manuscript has been to determine the significance of the direct pathway collaterals for behavior which, as discussed, has been technically challenging. We are grateful that the reviewer saw this and found that we have done this in a well-executed and elegant way. We agree that while our experiments support the hypothesis that pallidostriatal neurons contribute to the behavior we cannot exclude that other pathways are also participating. We further agree that this is a point that needs to be discussed in more detail, which we have done in the resubmitted manuscript (see our answers to questions 1, 2 and 3 below for more details).

1. I found this paper very well written yet, in this work, the authors associate Npas1+ GPe neurons with FoxP2+ arkyallidal neurons. For example, the authors state that ‘Npas1 in the GPe is almost exclusively expressed in arkyallidal neurons’. This concept is not accurate and sends a confusing message to the field. Indeed, arkyallidal FoxP2+ neurons only represent a subclass of Npas1+ cells. In fact, as already described in multiple studies, many Npas1+ neurons are not arkyallidal neurons with proportions ranging from 40% (see Dodson et al., PMID: 25843402; Cui et al., PMID:33731450) to 50% (see Abrahao et al., PMID: 29917235). This discrepancy between Npas1+ and arkyallidal neurons should be clearly stated and the manuscript revised accordingly.

We apologize for the confusion and agree with the reviewer that this point needs to be clarified. When introducing Npas1-cre mice we have now indicated that a subset of Npas1 neurons also project to cortex, thalamus and the midbrain and that only 55-70% of Npas1 neurons are truly arkyallidal (Abrahao et al. PMID: 29917235 shows that 55-60% of Npas1 neurons co-express FoxP2 in mice; Dodson et al. PMID: 25843402 find 56% in mice, Cui et al. PMID:33731450 find 57% in mice and Abdi et al. PMID 25926446 find 71% in rats). We had to use Npas1-cre mice instead of Foxp2-cre mice as Foxp2 is also expressed in the striatum, where we needed to express actuators under the *Drd1*-cre promoter. We updated the manuscript accordingly (page 14; line 29):

What could be the circuit mechanisms by which dSPN GPe terminals support motor function in the GPe? The GPe can be divided into two principal neuron classes, arkyallidal neurons (which all express FoxP2, most of which express Npas1; and which heavily project to the striatum) and prototypical neurons (which express Nkx2.1, most of which express parvalbumin, PV; and project to the midbrain) 16,20,22,23,61–63. We here hypothesized that dSPN “go” axons support motor function by functionally inhibiting ongoing motor-related activity in arkyallidal “stop” neurons 22,23. Indeed, recent work used Npas1-cre and PV-cre mice to label arkyallidal and prototypical neurons, respectively: they found that although PV+ neurons receive limited dSPN input and their activation promotes locomotion, Npas1+ neurons receive strong dSPN input and their activation suppresses locomotion 64–66. Similarly

in the in vivo anesthetized state 17,19, dSPNs were shown to strongly inhibit FoxP2+ neurons but only weakly inhibit Nkx2.1+ neurons. Therefore in a first step we set to determine whether the dSPN to arkyplallidal connection is active during ongoing motor behavior. Indeed it is important to verify this since recent work showed that active circuits in the GPe cannot always be predicted from in vivo physiology experiments done in the anesthetized state due to the presence of dense multisynaptic inhibitory circuits which can override monosynaptic connections to arkyplallidal cells 15. We here used *Drd1-cre* mice crossed to *Npas1-cre* mice. In mice or rats only 55-70% of *Npas1* neurons are considered arkyplallidal (the other 30-45% project to cortex, reticular thalamus and midbrain instead of striatum) 20,61–64; thus *FoxP2-cre* mice would be more selective for arkyplallidal cells than *Npas1-cre* mice. However, *FoxP2* is also expressed in the striatum which would interfere with our dSPN manipulation, while *Npas1-cre* selectively expresses in the GPe (see Fig. 1 in 67).

We also added a detailed discussion about the complexity of GPe neuron populations, their projections and that non-arkyplallidal *Npas1* projections may contribute to the behavioral effects of inhibiting direct pathway collaterals in the GPe (page 19; line 23).

We should address some of the limitations of our work including the possible implication of other subcircuits or other cell types in the GPe. Originally, GPe neurons have been divided into two populations, prototypical neurons and arkyplallidal neurons which differ by their firing patterns, projections and molecular markers (reviewed in 76). One proposed organization has been to oppose neurons expressing *Nkx2.1* which label prototypical cells to neurons expressing *FoxP2* which label arkyplallidal cells, but newer molecular classifications have also been used 16,20,22,23,61,62,76. To label arkyplallidal cells, we here chose to focus on the *Npas1* marker as *Npas1+* neurons were shown to receive strong input from dSPNs 66, suppress locomotion 15,65, and as we show here recapitulate the effects of bridging collaterals. A major advantage of this approach is that unlike *FoxP2-cre* mice, *Npas1-cre* mice only express cre in the GPe and not the striatum which allowed us to cross them with *Drd1-cre* mice for dSPN manipulations. However, while about 55-70% of *Npas1* neurons co-express *FoxP2+* 20,61,62,64, and are considered arkyplallidal neurons projecting to the striatum 61, 30-45% of *Npas1* neurons are *FoxP2*-negative but *Nkx2.1+* and *Lhx6+* and project to reticular thalamus, cortex and substantia nigra (mostly compacta) 62,67,77,78. Hence these extra-striatal *Npas1* circuits could also contribute to the behavioral effects of bridging collaterals. Furthermore, the contribution of prototypical cells remains to be clarified: indeed, recent work showed that dSPN input to PV neurons (which label most prototypical cells 76) is weak 16 and that PV neurons inhibit locomotion 65. Thus if dSPN bridging collaterals would act via inhibiting PV neurons, PV neurons should decrease and not enhance locomotion. On the other hand, a recent paper showed that bridging collaterals dSPNs target a small population of *Lhx6*-negative parafascicular thalamus-projecting PV cells, but not STN/SNr-projecting PV cells, which regulate reversal learning, but not locomotion. Moreover, a small population of *Lhx6+* cells that are negative for PV and *Npas1* 62,67 have not been tested as to whether they receive dSPN input and their role in motor control is unknown. Finally, we observed here that ChAT neurons, which are targeted by dSPNs 68, do not recapitulate the motor effects of bridging collaterals. It is possible, however, that ChAT neurons mediate other behavioral effects of dSPN collaterals not addressed here.

Of note, our optogenetic inhibition studies in anesthetized mice could suggest that dSPNs inhibit more than the 30% cells formed by *Npas1* neurons (plus about 5% of ChAT neurons) in the GPe since depending on the analysis method we observed inhibition in 30-50% of units. However, inhibited units may also arise from polysynaptic rather than monosynaptic inhibition which we cannot distinguish here (since we measure the effect of inhibition of neurons that are active at 10-40 Hz frequency). Polysynaptic effects may explain why in our previous study⁶ activation of iSPNs with the

same Chr2 setup led to inhibition of 70% of units, even though iSPNs should primarily target Nkx2.1+ or PV+ cells which represent only 40-60% of the GPe population^{20,61,63,67,77,79}. Last, there are Drd1/Drd2 co-expressing neurons in the DMS that project exclusively to the GPe²⁶. These neurons should be targeted by our approach using Drd1-cre mice. However, they only represent a small % of SPNs, and do not promote locomotion like bridging collaterals do²⁶. Thus, while our data are consistent with arky pallidal neurons mediating the effects of bridging collaterals on motor function future studies will have to address the involvement of other GPe subcircuits in this effect.

2. My previous comment leads to a second consideration regarding the circuit mechanism that might underlie the functional effect observed after inhibition of the dSPNs bridging collaterals in the GPe. Currently, the authors interpret their results through the contribution of pallido striatal circuits. This is partly justified by the fact that all Npas1+ neurons display pallido striatal projections. However, as mentioned above, a substantial proportion of these neurons are non-arkypallidal Npas1+ neurons that also send projections to classic targets of GPe neurons (such as the STN and the SNr), but also to unconventional targets (such as the cortex, see Abecassis et al., PMID: 31811030). The functional impact and contribution of all these different projections formed by Npas1+ neurons are not considered in this work. This represents an important limitation that should at least be acknowledged and thoroughly discussed in the manuscript.

We agree with the reviewer and added a detailed discussion about the complexity of GPe subpopulations including Npas1 neurons that project to cortex, thalamic reticular nucleus and substantia nigra (mostly compacta). These additional projections could contribute to the behavioral effects. Of note it was previously reported that within the substantia nigra Npas1 neurons project more heavily to SNc than SNr (Abecassis et al. PMID: 31811030). Future studies should address whether these extra striatal Npas1 projections contribute to the behavioral effects of bridging collaterals (See our response to question 1; and Page 19; Line 23).

3. The authors mentioned that dSPN inputs inhibit Npas1+ or FoxP2+ arky pallidal neurons and cite three references to support this claim (Cui et al., 2021; Ketzef & Silberberg 2021; Johansson & Ketzef, 2023). However, work by Spix et al., (PMID: 34618556) has also described that these dSPN projections are not exclusive to arky pallidal neurons but also drive 'robust inhibition in Lhx6-GPe neurons' (which are STN/SNr downstream projecting neurons). Once again, this is problematic as it would suggest that the effect of dSPNs collaterals in the GPe could also be relayed by other means than pure pallido striatal circuits. It would be nice to discard these potential 'non-pallido striatal' effects (or at least mention their existence as potential drawbacks).

About half (or less) of Lhx6+ cells in the GPe are PV+ and half (or less) are Npas1+ (PMID 26311767, 29917235). It is our understanding that Lhx6-GPe neurons which could potentially receive direct pathway input (suggested in Spix et al. PMID 34618556 but the study was not designed to determine monosynaptic inhibition) are largely Npas1+ neurons since direct pathway input to Npas1 neurons is strong whereas input to PV cells is weak (Cui et al., PMID 33731445 and Spix et al. PMID 34618556). This Lhx6+-Npas1+ population is nearly identical to Nkx2.1+Npas1+ cells that do not project monosynaptically to the STN, SNr but rather to cortex, thalamus and SNc (PMID 26311767, 31811030, 29917235). A small proportion of Lhx6+ cells are PV-, Npas1- (PMID 26311767, 29917235) and could potentially also receive direct pathway input (and contribute to the findings in Spix et al. PMID 34618556); but this was not examined yet. As indicated above we now discuss how the effects of dSPN collaterals on cortex, thalamus and SNc projections could contribute to the behavioral effects (page 19; line 23).

While it would be great to show this experimentally, further dissecting the GPe downstream pathways in the regulation of behavior would be beyond the scope of this manuscript. We do agree with the reviewer that we should discuss this in more detail in the revised manuscript.

4. In agreement with my previous comment, figure 6G shows that a high proportion of GPe neurons (close to 50% of the recorded cells in the saline condition) are inhibited by dSPN optogenetic stimulation. This goes against dSPNs collaterals only impacting arky pallidal neurons (that represent less than 30% of all GPe neurons). Also, here the authors only make the distinction between 'inhibited vs. non-inhibited cells'. What do 'non-inhibited cells' include as neuronal responses? For example, did the authors record cells that were excited by dSPN optogenetic stimulation?

High proportion of inhibited neurons: A high proportion of GPe neurons (55%) are detected as significantly inhibited using our initial normalized spike frequency analysis: this is a good point that requires discussion. Note that we now included a new (more conservative analysis) using Z-scores (see below) in which we detect that 33% of neurons are inhibited. How can we interpret these numbers? It is possible that dSPNs inhibit other neurons than Npas1 and ChAT (which represent 30% of cells and 5% of cells, respectively). We now detail this above (see answer to questions 1,2,3). However, it could also be the case that some of the inhibited units arise from polysynaptic rather than monosynaptic inhibition which we cannot distinguish here (since we measure the effect of inhibition of neurons that are active at 10-40 Hz frequency). Polysynaptic effects may explain why in Cazorla et al. (PMID 24411738) activation of indirect MSNs with the same ChR2 setup led to inhibition of 70% of units, even though iMSNs should only target 40-50% of the GPe population (PMID 26905595, 26311767, 31811030, 25926446). We included a discussion of this point in Page 11; Line 27 and Page 19; Line 23 (s.above).

An average of 55% (considering normalized spike frequencies, see Fig. 6G) or 33% (considering Z-scores, see Fig. S6C) of GPe neurons were inhibited, which could be due to a mix of monosynaptic and polysynaptic effects since dSPNs are thought to target Npas1 (~30% of GPe cells) and ChAT cells (5%)¹⁵⁻¹⁹.

Inhibited cells: Non-Inhibited units include all cells that were not considered inhibited (this includes cells that did not change significantly, or could include the low number of excited units; see below). In Fig. 6, inhibited units were neurons that decreased their response by 1/3 of the pre-stimulation firing activity. We now also add additional analyses (**Supplementary Fig. S6**) using a Z-score based analysis which provided a more stringent criterion for identifying units whose activity was decreased by the optogenetic inhibition. Here Z-score of spike frequency was calculated as: $[(\text{stim period mean}) - (\text{baseline period mean})] / (\text{baseline period SD})$. A Z-score of -2 and +2 was used as the cut-off value to define cells that are significantly inhibited or excited, respectively. Overall, our Z-score analyses reveal the same conclusion as the Normalized spike frequency analyses, namely that CNO in the GPe blunts opto-induced inhibition of GPe units but not in the SNr (see **Revision Letter, Figure 1**).

Figure 1: Z-score analysis of inhibitory responses. C. Proportion of GPe units for which the Z-score of the spike frequency is significantly decreased (inhibited units) or not (non-inhibited units) after laser stimulation (Fisher's test SAL vs CNO: at 0ms $p = 0.99$; at 250, 500 ms: $***p < 0.001$; at 1000 ms: $**p = 0.0024$). D. Same as C. for the SNr (all SAL vs CNO: $p = 0.99$). N= 5/group.

Excited cells: We did record excited units following dSPN optogenetic stimulation but as expected the number of excited units was very low (0 to 8 neurons out of 45 to 50 neurons). We now report this new analysis in **Supplementary Fig. S7**. There are no clear effects of SAL/CNO treatment; however since the number of excited units is so low across conditions, we cannot perform statistical comparisons (see **Revision Letter, Figure 2**).

Figure 2. Normalized spike frequency analysis of excitatory responses. A. Proportion of units in the GPe for which the normalized spike frequency is significantly increased (excited units) or not (non-excited units) after laser stimulation. The actual number of excited units found is labeled in the bar graphs in parenthesis. B. Same as A for the SNr. The actual number of excited units found is labeled in the bar graphs in parenthesis. Z-score analysis was also performed and shown in **Supplementary Fig. S7**.

5. As stated before, one limitation of this work is the lack of a precise circuit mechanism to explain the behavioral effect caused by dSPNs bridging collaterals inhibition in the GPe. In particular, the absence of supportive evidence in favor of a pallidostriatal circuit weakens the authors' conclusion. If time allowed, I believe that this could easily be addressed by a

straightforward experiment showing that the striatal activity (measured through GCaMP signal in dSPNs) was truly altered during motor behavior following the inhibition of dSPNs bridging collaterals in the GPe.

In support of a contribution of the pallido-striatal pathway we show that direct pathway collaterals inhibit GPe Npas1 neurons during behavior and that Npas1 neurons, which in large part project back to the striatum regulate motor behavior. The proposed experiment would further strengthen this evidence by showing that the disinhibition of Npas1 neurons would inhibit dSPN activity during behavior. This is a great experiment but unfortunately not straightforward due to technical reasons. In order to do the experiment eOPN3 or hM4D would have to be selectively activated in the GPe and striatal dSPN activity measured. Due to the limited space, one of the optic fibers (or cannula) would have to be inserted with an angle. Doing surgeries involving angles and 2 fibers very close to each other are very difficult to do and we don't have this set up in the lab yet. Furthermore, dSPNs and Npas1 are silent at rest, inhibit each other, and Npas1 inputs to the striatum are weak, therefore we would need to find the right conditions so that the cells are active enough allowing us to detect inhibition. Altogether, we estimate that this could easily take 1 year to perform and although very interesting, would end up being beyond the scope of this paper. Accordingly, we agree that we need to emphasize that other mechanisms may contribute to the behavioral effects, so as to not confuse the field. We have now added a detailed discussion in the revised manuscript that Npas1 projections to other brain regions, and Lhx6+-Npas1-,PV- neurons could also contribute to the effects of dSPNs bridging collaterals. These edits can be found Page 19; Line 23 (and in response to comment 1).

6. The open field motor classification performed in Fig.7 E, F includes locomotion frames, motionless frames, and a category defined as 'fine movements' for video frames that did not fall into the 2 first categories. However, looking at the videos, I really wonder if the video's quality (and field of view) was sufficient to provide any information on fine movements. In addition, does this 'fine movements' category contain heterogeneous motor behaviors? If yes, wouldn't it fall better as a 'background/noise' category?

We agree with the reviewer that the term 'fine movements' is confusing. We now term this "non-locomotor movements" and indicate in the revised manuscript that this could include, but is not restricted to, fine movements e.g. head movements, rearing, grooming, etc.

Results; page 14; line 20: Trajectories of mouse body parts (obtained with DeepLabCut) were used to classify frames into 3 categories: locomotion, motionless and non-locomotor movements (this includes but are not restricted to head movements, rearing and grooming)

Figure 7, Figure Legend: E. Left: Heatmaps showing behavioral classification of videoframes (all mice) into locomotion (body center speed >4.5cm/s), motionless (speed of all body parts ≤0.8cm/s) or other non-locomotor movements (does not fulfil locomotion or motionless criteria). Non-locomotor movements include but are not restricted to head movements, rearing, grooming and other fine movements. Note the mild locomotion increase 5 sec before laser onset (dashed line).

Methods: Locomotion frames were defined as frames when body speed reached >4.5cm/s. Motionless frames were defined as frames when the speed of all body parts was ≤ 0.8cm/s. Other non-locomotor frames were defined as frames not falling into the other categories.

7. In the discussion p18/29, the sentence line 16-17: 'Moreover, recent work showed that iSPN opto-stimulation disinhibits Npas1 neurons in vivo via a disynaptic circuit' is citing the ref: Cui et al., 2021. However, please note that the cited paper does not contain in vivo recordings, nor does it reveal a disynaptic disinhibition of Npas1+ neurons. In fact, this work clearly shows (see Fig. 7d) and states the opposite: 'On the other hand, although there was a big difference in the strength of the DLS-iSPN input between PV+ neurons and Npas1+ neurons, we did not observe a difference (p=0.083) in the fold change of firing between PV+ neurons and Npas1+ neurons from DLS-iSPN stimulation...'. Please amend the text accordingly

Thanks for catching this mistake. We cited the wrong paper, it was supposed to be Aristieta et al., PMID: 33306949, in which authors show that iSPN opto-stimulation disinhibits arky pallidal neurons *in vivo* via a disynaptic circuit. We have corrected this in the resubmission (page 19; line 21).

8. The authors reported the dilution of all the viral solutions used in this manuscript but, for best reproducibility, the stock titer should also be provided.

Stock titers and dilutions were provided at submission in **Supplementary Table 2** as a separate file. We noticed that it was not clear what the titers corresponded to (i.e., diluted or stock). We edited this in the revised version. We also now use the labels “undiluted” or “diluted 1:1 AAV:PBS” since we think this formulation is more clear. Further, we moved this data to the Supplementary Material word document and include it as an embedded Table (still as **Supplementary Table 2**) so that they are easier to find for the reader.

9. I could not find the stereotaxic coordinates used by the authors for all their experiments. I might have missed it but, if not, it would be nice to provide them directly in the methods.

Stereotaxic coordinates were in a separate Excel sheet document called **Supplementary Table 1**. We moved it to the Supplementary Material word document and include it as an embedded Table so that they are easier to find for the reader.

Reviewer #2:

Here, Labouesse and colleagues performed a variety of experiments to test the functionality of “bridging” collaterals from direct pathway striatal projection neurons (SPNs) to the GPe in awake behaving mice. Specifically, a series of photometry, chemogenetic, and optogenetic experiments were carried out. These experiments potentially support two important phenomena: (1) the dSPNGPe projection is relevant for motor behavior and (2) the effect of this projection is primarily mediated by Npas1 GPe neurons. The number of experiments and amount of work here is very impressive; however, I found the presentation confusing to follow, lacking some key details, and I did not fully understand a few essential controls. I am hoping that my confusion can be resolved through careful reanalysis of the data rather than carrying out any new experiments. If this can be resolved, then I see path for this manuscript being suitable for publication in Nature Communications.

We would like to thank the reviewer for the overall positive assessment. We hope that the resubmission will resolve the remaining confusion.

1. Use of the word collateral: Can the authors clarify a few basic points re: bridging collaterals? After reading through this manuscript and cited works I’m confused about whether *all* dSPNGPe projections are in fact axon collaterals.

and

Secondly, and more importantly, is there evidence of dSPNGPe projections that are not axon collaterals? Some of the cells in Figure 1E look like they express HSV and are not labeled by retrobeads – admittedly the picture is fuzzy in the PDF so maybe I’m missing something here. There is no quantification so I’m left wondering if there is a non-trivial number of dSPNs that project to GPe and not SNr .

Figure 3: Sparse labelling of direct pathway collaterals in Fig. S2 of Cazorla et al. *Neuron* 2014

This is a very good point. If we follow the definition, the direct pathway projection should project to the SNr and/or EP. To our knowledge SPNs have not been shown to have two independent axons. From this it would follow that GPe terminals must arise from collaterals. This is in line with single neuron labelling studies showing that the majority of SNr- or EP-projecting striatal neurons have collaterals in the GPe. In the rat, of about 120 striatal projection neurons that have been individually labeled in three independent publications (PMID 21314848, 1698947, 10997578), 37% projected exclusively to the GPe (“pure” indirect pathway), whereas only 3% projected only to the SNr or EP (“pure” direct pathway). 60% of labeled neurons projected to the SNr/EP and possessed collateral terminal fields in the GPe. This posits that the large majority of direct pathway (SNr projecting) neurons have axon collaterals in the GPe. In our own hands we traced some of the collaterals in Cazorla et al. 2014 (PMID 24411738) using a low titer Cre dependent AAV expressing ChR2 as an anatomical tracer in *Drd1-cre* mice (**Revision Letter, Figure 3**). In our sparse labelling we did not detect any terminals that were not connected to an axon going through the structure. This would indicate that the large majority or all dSPN→GPe projections arise from collaterals. However, our sparse labelling study at the time was not designed to address the question of collaterals vs. main terminal fields and we may have

overlooked some limited *Drd1-cre* SPN projections that only target the GPe.

To determine the possible existence of dSPN→GPe cells that do not arise from axon collaterals, we now used our HSV-tracing to quantify cells; we find that out of 159 YFP+ cells, 147 were also retrobeads+ (92.5%) and 12 were retrobeads- (7.5%). In the same area, out of 148 retrobeads+ cells counted, all but 1 were YFP+ (99.3%). This indicates that the very large majority (>99%) of SNr-projecting dSPNs have collaterals in the GPe consistent with the single neuron labelling studies cited above. In addition, out of all GPe-projecting dSPNs, a small population (7.5%) does not project to the SNr, and thus do not arise from axon collaterals. This population possibly corresponds to DMS cells described in Bonnavion et al., *bioRxiv* 2023; <https://doi.org/10.1101/2022.04.05.487163>) which are *Drd1+/Drd2+* co-expressing cells, represent a small % of SPNs (6-16%), and do not promote locomotion like bridging collaterals do. Importantly, in the *Drd1-cre* based targeting approach in this paper, we may be targeting this small (7.5%) population of GPe-projecting *Drd1* cells that do not project to the SNr.

We hope this clarifies the reviewers point. We added a description about the single neuron labeling studies to the introduction (Page 3, Line 15), added the quantification to the manuscript (Page 4, line 4) and discussed the impact of targeting the limited D1/D2 population in the discussion (Page 20, line 7).

Single neuron labelling studies performed in over 100 projection neurons in the rat have shown that 37% projected exclusively to the GPe (“pure” indirect pathway), whereas only 3% projected solely to the SNr or EP (“pure” direct pathway). 60% of labeled neurons projected to the SNr/EP and possessed collateral terminal fields in the GPe ¹²⁻¹⁴.

Quantification showed that out of 159 YFP+ cells, 147 were also retrobeads+ (92.5%) and 12 were retrobeads- (7.5%). Out of 148 retrobeads+ cells counted, all but 1 were YFP+ (99.3%). This indicates that the majority (99%) of SNr-projecting dSPNs have collaterals in the GPe consistent with ¹²⁻¹⁴. In addition, out of all GPe-projecting dSPNs, a small population (7.5%) does not project to the SNr, possibly corresponding to cells described in ⁷⁸.

Last, there are *Drd1/Drd2* co-expressing neurons in the DMS that project exclusively to the GPe. These neurons should be targeted by our approach using *Drd1-Cre* mice. However, they only represent a small % of SPNs, and do not promote locomotion like bridging collaterals do ⁷⁸.

First a minor point: in Figure 1d, assuming I understand it correctly, these numbers are substantially larger than the prior report in Cazorla et al. (Fig. 1e). Can the authors comment on this?

Good point: In Cazorla et al. (PMID 24411738), we quantified the density of terminal fields in the GPe and SNr (% of striatum) using IHC against GFP as a readout arising from a *Drd1a-GFP* BAC transgene. Thus, the signal originated from GFP on both axons and terminals in the GPe/SNr and from cell bodies, dendrites, axons and terminals in the striatum. Here, we used IHC against Synaptophysin-GCaMP6s expressed with a Cre-dependent AAV in *Drd1a-Cre* mice. Thus, signal intensity arises primarily from terminals with much less contribution of axons, dendrites and cell bodies. In Cazorla et

al. the GPe and SNr densities were 45% and 105% of SNr density, whereas in the submitted paper the GPe and SNr densities were 80% and 140% of striatal density. In Cazorla et al., both the nominator (GPe/SNr) and the denominator (striatum) should be affected by the non-selective expression of GFP. In particular, not targeting terminals should lead to an overall increase in intensity in the striatum (denominator) since it also includes expression in cell bodies and dendrites; leading to overall smaller reported numbers of % expression. We indicated in the Methods section (page 23; line 5) the difference between the two methods, which may explain the difference.

Note that here GFP expression is targeted to the pre-synapse thanks to the Synaptophysin construct, whereas in Cazorla et al. 2014⁶, GFP was unspecifically expressed in dendrites/cell bodies in the striatum and axons in the projection fields. This explains the differences in GPe/SNr % of terminal field (arguably the present quantification is more accurate).

Also, if indeed some cells appear to only project to GPe, an alternative explanation is that some iSPNs are labeled due to viral expression leak – I do not know what the likelihood of this is with the HSV used by the authors. It's doubly hard to assess the likelihood of the latter possibility since the viral titers are not listed in the methods section (note that the same concern is there for AAVs due to not listing titers). I'm hoping this is a misunderstanding on my part and new experiments are not needed. Either way, this needs to be shored up since the author's interpretations rest on the idea that all dSPNGPe terminals come from axon collaterals.

Another important point is the concern of leaky non-cre dependent expression of the HSV that would label dStr->GPe cells non-specifically. To examine this we injected the HSV virus into Drd1-cre negative mice and verified lack of expression. As shown in **Revision Letter, Figure 4** below, we could not detect fluorescence expression in Drd1-cre negative mice. Furthermore, in Drd1-cre positive mice, the HSV tracer (labeled with YFP) colocalized with cre; no YFP cells was cre-negative, and no YFP or cre expression or colocalization was detected in Drd1-cre negative mice. This indicated that the tracer only marks Drd1-cre cells. Thus, with this new dataset we can exclude the possibility that some iSPNs are labeled with HSV due to viral expression leak. The new figure is provided in the revised manuscript as **Supplementary Figure 3**.

Figure 4. Validation of retrograde tracing strategy to trace dSPN projections from dStr to GPe. Representative images showing colocalization of YFP (cyan) with cre (pink) in the striatum of Drd1-cre positive mice (white arrows) injected with rgHSV-YFP in the GPe (left). No YFP+, cre- cells were detected in Drd1-cre positive mice. No YFP or cre fluorescence was detected in Drd1-cre negative mice injected with rgHSV-YFP in the GPe (right). This indicates that the retrograde tracer only recombines in cre-positive cells.

2. Viruses used and signal conditioning: The authors convincingly show that terminal calcium signals in GPe and SNr are correlated – of course this should be the case if both originate from the same dSPNs. However, I'm concerned about a few things here that might be over-stating the result: (1) there is not a direct comparison between the synjGCaMP8s results and the jCaMP7s results on the rotarod (2) I could not find a characterization of synjGCaMP8s kinetics in the manuscript, and (3) the signals all arise from slow GCaMP variants and signals are heavily smoothed after digitization.

First, as the authors openly admit, Figures 2 and 3 arise from non-terminal targeted jGCaMP7s. This motivates them to develop a synGCaMP based on jGCaMP8s, which is used in Figure 4 for a second stab on rotarod recordings. Rather than showing results in a separate figure, it would be more useful to directly quantify how much of the Figure 3 results are due to cross-talk from dSPN->SNr axons. By eye it looks like there's a 15% reduction in Pearson correlation when targeting to terminals (i.e. it's a significant, but not huge contribution, which is reasonable). It is important to note that synGCaMP targeting is likely not perfect (as the author show), and there is still probably a small contribution from dSPNSNr axons

Fig. 3 (GCaMP7s) originally showed the data acquired by the photometry processor, while Fig. 4 (SyGCaMP8s) data was low-pass filtered offline at 1Hz. To stay consistent, Fig. 4 is now also showing acquired (non-filtered) data. When comparing Fig. 3 with Fig. 4 (**Revision Letter, Figure 7**), we calculated the following Pearson r correlations: GCAMP7s vs SyGCaMP8s rotarod off: 0.90 vs 0.69, i.e. a 23% decrease; GCAMP7s vs SyGCaMP8s rotarod on: 0.70 vs 0.47, i.e. a 33% decrease

vs. GCaMP7s. This suggests that 23-33% of the Pearson r correlation in GCaMP7s could be due to calcium signals in axons. We discuss this now in the results section (page 10, Line 9):

Since Pearson r values obtained with jGCaMP7s were 20-30% higher than with synaptophysin-jGCaMP8s this suggests that 20-30% of the correlation in the jCaMP7s could be due to Ca^{2+} signals in axons. It is likely though that even with the terminal-targeted SyGCaMP8s there may still be a small contribution of axonal signal. Moreover, because jGCaMP8s has faster on and off kinetics than jGCaMP7s (half rise-time: 21 vs. 67 ms; half-decay time: 52 vs. 81 ms⁵⁶. Note: SyGCaMP8s kinetics may slightly differ vs. jGCaMP8s), differences in the sensor kinetics could contribute to the differences in the degree of correlations. Regardless of the source of variation, we find that results obtained with both sensors concur, namely that correlated activity is higher in the rotarod off vs. on condition. This confirms our finding that dSPN cell bodies send shared information to the terminals in GPe and SNr, but that there are additional local factors regulating terminal activity in a task-dependent manner.

We also should add that the two sensors are not exactly the same (e.g. one is 7s, one is 8s) and other factors may be at play that could explain these differences (see next comment), thus we are thinking it might be better to not directly compare the sensors in the same figure.

3. Viruses used and signal conditioning:

Second, there is no characterization of synjGCaMP8s kinetics. A good first step here is to report the autocorrelation in baseline data from all GCaMP variants (and targets where it was expressed) used in the manuscript. If synjGCaMP8s has substantially slower kinetics, then it's possible that synjGCaMP8s is under-reporting the difference between GPe and SNr terminal photometry signals.

It is indeed interesting to consider kinetics in this context. Between submission of this paper and now, the GCaMP8 paper was published. This paper indicates that if anything, jGCaMP8s kinetics is faster: in cultured neurons jGCaMP8s has substantially faster on and off kinetics (half rise-time: 21 ms; half-decay time: 52 ms) than jGCaMP7s (half rise-time: 67 ms; half-decay time: 81 ms) (PMID 36922596). Indeed, jGCaMP8s is actually faster than jGCaMP7f (though it is slower than jGCaMP8f). We choose 8s vs 8f because Ca^{2+} transients in terminals are small and we needed the most sensitive sensor. Since we added a targeting motif (synaptophysin) to the GCaMP8s, the kinetics may be slightly different. To address this, we now also analyzed the autocorrelation of the in vivo calcium signal for jGCaMP8s and GCaMP7s as a proxy for kinetics. An autocorrelation that quickly decreases when increasing the lag (seconds) potentially reflects a signal with more "fast" fluctuations (potentially meaningful, potentially noise), and in principle, faster kinetics. This analysis was done at baseline (selected as epochs when animal speed in an open field was $<3\text{cm/s}$). We find that the autocorrelation for SyGCaMP8s decreased faster at increasing lags compared to GCaMP7s; this was evident in the GPe (not SNr) likely due to the fact that there are less synapses and therefore lower signal to noise ratio. This is consistent with the faster kinetics of jGCaMP8s. Therefore, it is unlikely that SyGCaMP8s is underreporting the difference between GPe and SNr terminal photometry signals. We now added information about kinetics in the revised paper: (page 10, Line 3 and 10):

A. Auto-correlation of raw data: GCaMP7s vs. SyGCaMP8s

Figure 5. Autocorrelation analysis of GCaMP7s and SyGCaMP8s done data acquired with the photometry processor (3Hz low-pass online; no offline low-pass filter)

We chose jGCaMP8s due its superior sensitivity (1 AP dFF 9.21 vs. GCaMP7s 4.95).

jGCaMP8s has faster on and off kinetics than jGCaMP7s (half rise-time: 21 vs 67 ms; half-decay time: 52 vs 81 ms⁵⁵),

4. Viruses used and signal conditioning:

Third, I'm hoping the authors can re-analyze a subset of the data with much less smoothing (see a related minor point below). The slow variants of GCaMP are already smoothing calcium activity (note: repeating these experiments with different

GCaMPs is overkill in my opinion), so it is important to know whether the post-hoc smoothing is masking fast time-scale differences.

This is a very good point. We looked again at this data and noticed that we did not provide sufficient detail in the Methods with respect to what data was “raw” and what data was low-pass filtered. Initially Figure 2 (GCaMP7s, open field) and Figure 4 (SyGCaMP8s, rotarod) was presented as filtered data (low pass filter 1Hz), while Figure 3 (GCaMP7s, rotarod) was presented as raw data (= no offline low-pass filter). The low-pass filter was included since behavioral changes of interest were slower than 1Hz (see also our answer to question 12 below). Of note, this low-pass filter did not induce alterations in the data (see e.g. **Revision Letter, Figure 6**) or affect the results qualitatively (see e.g. **Revision Letter, Figure 7**). To be more consistent, we now present all Figures 2, 3, and 4 with the raw data (i.e. no offline low-pass filter at 1 Hz; the only low-pass filter applied was online and was at 3 Hz).

Consistent with what the Reviewer predicts, the Pearson correlation for SyGCaMP8s is smaller when the data is not low-pass filtered (**Revision Letter, Figure 7** and Manuscript **Fig. 4**). Importantly, the difference between on/off epochs does not qualitatively change, nor do the conclusions for the paper.

Now that both GCaMP7s and SyGCaMP8s rotarod Pearson r correlations between GPe and SNr are presented in the ‘raw data’ form, it is easier to make comparisons. As one can see (**Revision Letter, Figure 7**), the Pearson coefficients with GCaMP7s are overall higher (see also our answer to the previous question). This difference is likely to be due partly to faster kinetics for SyGCaMP8s, partly to axon contribution in GCaMP7s which increase the correlations. What is important to note, is that our conclusions remain the same, namely that the GPe/SNr correlation is generally high and that the correlation decreases when the rotarod is on (task-dependent).

Of note, the only GCaMP data where we kept an offline filter (smoothing in 0.5 s moving average window) is shown in **Supplementary Figure S4D-E** where we quantify peak frequency, amplitude or interpeak interval. Here it was important to apply a filter to avoid overcounting high-frequency peaks that poorly track the behavior of interest (jumps). For instance, one can see in **Fig. 3F** that relevant behavioral transients (jumps) are 0.5 to 1.5 sec long. This is consistent with data computed in **Supplementary Figure S4D-E** where for example we count interpeak intervals of 0.5 to 4 sec. Of note this smoothing does not alter the data (**Revision Letter, Figure 8**). We now added directly in the Figure Legend of **Supplementary Figure S4D-E** that data was smoothed to avoid any confusion.

A. SyGCaMP8s: Comparison raw data vs. data filtered with a 1Hz low-pass filter

Figure 6. Comparison between raw and low-pass filtered (1Hz) $\Delta F/F$ fluorescence data obtained with SyGCaMP8s. A. Raw (blue) and filtered (red) data overlay.

Figure 7. Comparison of GPe/SNr Pearson correlation coefficients between raw data collected with GCaMP7s or SyGCaMP8s and filtered data (1Hz) collected with SyGCaMP8s. The raw data is shown in the revised paper.

A. GCaMP7s: Comparison raw data vs. data filtered with a 0.5s moving average smoothing

B. GCaMP7s: Peak detection with raw data vs. with a 0.5s moving average smoothing

Figure 8. Comparison between raw and 0.5s smoothed data (in a moving window) $\Delta F/F$ fluorescence data obtained with GCaMP7s. A. Raw (blue) and filtered (red) data overlay B. Raw (blue) and filtered (red) data peak detection.

Methods: For peak frequency and amplitude calculation, dFF was low-pass filtered using a 0.5 s moving average smoothing to avoid overcounting peaks not relevant to the behavior of interest (jumps).

5. Controls for chemogenetic manipulations:

I am confused about the controls for a key experiment – chemogenetic inhibition at dSPN axon terminals in GPe. Ideally CNO does not directly impact the dSPN→SNr projection. It is unclear if this is just an unrepresentative example, but the baseline firing rate of the neuron shown in Fig. 6F (CNO) appears substantially lower than the neuron in 6E (SAL). Additionally, the opto-response looks fairly blunted, especially in the early phase in response to the 250 ms laser pulse. I recommend a careful reanalysis of this dataset. Is there a systematic shift in the baseline firing rate? I am also wondering if the normalization shown on the right side of Fig. 6 masks the blunting of the opto-response. How do these results look when analyzing the raw difference in firing rate due to opto-stim? I could not find details on calculation the normalized response, but what if the normalized response is measured by taking $\min(\text{firing_rate})$ during stim and comparing to baseline? I wonder if some of the subtle effects are washed out by summing or averaging.

Baseline activity CNO vs saline: This is a good point. Indeed, it looks like that there is a decrease in the basal spike frequency of SNr neurons when comparing Fig. 6F with 6E. We now performed a repeated measures ANOVA to quantify this by comparing the 1 second pre-laser time window between saline and CNO for the different stimulation durations. Although visually there is a mild decrease in basal spike frequency, this does not seem systematic as mostly driven by 1 animal in the SAL condition, and is not close to significance (**Revision Letter, Figure 9**). Similarly, when only the saline condition is analyzed for the entire 3 seconds depicted in manuscript Fig. 6E/F we did not measure any significant difference (data not shown, $p=0.34$). Comparatively, with the same $n=5$ in this experiment we are able to detect a 1.5-2 fold change in firing rate in the GPe with CNO vs. SAL (Fig. 6I, $p<0.001$), and this is not driven by a singled-out outlier; indicating that our study is not underpowered. Basal spike frequency is now added as: **Supplementary Fig. S6A-B**.

Figure 9. Comparison of baseline activity of SNr neurons between CNO and saline conditions. No differences were measured, rmANOVA: stimulus duration x drug, $p=0.60$; stimulus duration, $p=0.12$; Drug, $p=0.37$.

Effect of CNO on light induced inhibition of GPe/SNr neurons: Normalized responses were calculated by dividing the mean activity during the stimulation epoch [250, 500 or 1000 ms (& 1000 ms for 0 ms condition)] by the mean of the baseline activity obtained during the 1000 ms preceding laser illumination. They were analyzed for all units and for the 'significantly' inhibited (and excited) units, separately. We updated this more detailed definition in the Methods section (page 25, line 3). Data were normalized to account for differences in baseline firing between different neurons.

Blunting of response in SNr? The reviewer is correct that visually CNO seems to blunt the opto-response in the SNr, especially in the first 50 msec (manuscript **Fig. 6E vs F**). We re-analyzed the data to determine whether we may have overlooked mild effects of CNO on direct pathway induced inhibition of the SNr. We first compared the normalized firing rate during the first 50 ms where the PSTH looks different by eye. We did not find any significant differences between CNO and saline condition (**Revision Letter, Figure 10**). This is now added to Fig. 6 (**Fig. 6J,L**).

L SNr, Normalized spike frequency, first 50ms of Laser

Figure 10. Normalized spike frequency rate in the SNr in the first 50 ms (ANOVA: stim duration x drug $p=0.47$, drug $p=0.96$, main effect of stim duration: $***p<0.001$; post-hoc all mice pooled: 0 ms vs. other stim durations: all $***p<0.001$).

We then used a Z-score based analysis which provided a more stringent criterion for identifying units whose activity was decreased by the optogenetic inhibition (below 2SDs versus a 30% decrease of AP frequency). Although the Z-score appears slightly decreased in the SNr with CNO, we could not detect any significant effects, either by looking at the entire stimulation epoch (ANOVA: stim duration x drug $p=0.26$; drug $p=0.25$) or the first 50 ms (ANOVA: stim duration x drug $p=0.28$; drug $p=0.63$) (**Revision Letter, Figure 11**). This is now added as: **Supplementary Fig. S6**. Last, we used both methods to identify excited neurons and depending on the

analysis (Z-score vs normalized firing) between 0 to 8 of units were excited (now added as: **Supplementary Fig. S7**). The numbers were too low for a statistical analysis to detect an effect of CNO.

Raw difference in firing vs. Z-scores? We did not analyze raw differences in firing rates because fast-firing neurons (even if they would be in the minority) would be over-represented and therefore mask any (significant/non-significant) effects in low-firing neurons (even if they were in the majority). To avoid this, we originally normalized the data. For the resubmission we added the transformation into z-scores because it will compare the pattern/distribution of SDs of the baseline and optogenetic responses. It is an additional common method to analyze in this context.

The new text related to this analysis can be found page 11, line 26.

Figure 11. Z-score of the spike frequency in the GPe calculated in the full laser epoch (F) or in the first 50 ms (H) for all units. F. ANOVA: stim duration x drug $p=0.26$; drug $p=0.25$, main effect of stim duration: $***p<0.001$; post-hoc all mice pooled: 0 ms vs. other stim durations: all $***p<0.001$. H. ANOVA: stim duration x drug $p=0.28$; drug $p=0.63$, main effect of stim duration: $***p<0.001$; post-hoc all mice pooled: 0 ms vs. other stim durations: all $***p<0.001$.

Consistent with previous work^{6,17}, in Saline control mice dSPN opto-stimulation led to an inhibition of spike firing frequency in the GPe (**Fig. 6C**) and SNr (**Fig. 6E**). An average of 55% (considering normalized spike frequencies, see **Fig. 6G**) or 33% (considering Z-scores, see **Fig. S6C**) of GPe neurons were inhibited, which could be due to a mix of monosynaptic and polysynaptic effects since dSPNs are thought to target Npas1 (~30% of GPe or cells) and ChAT cells (5%)^{15–19}. The dSPN opto-induced inhibition of GPe spike firing was blunted when dSPN GPe terminals were chemogenetically inhibited via local GPe CNO infusion (**Fig. 6D**). This confirmed that local GPe CNO infusion in hm4D-expressing Drd1-cre mice (**Fig. 5**) inhibits synaptic release at dSPN GPe terminals, in line with the established role of hm4D as a presynaptic release inhibitor⁵⁷. Importantly, local CNO infusion into the GPe did not affect opto-induced inhibition of SNr spike firing activity (**Fig. 6F**). Upon quantification, we found that local GPe CNO infusion significantly reduced the number of inhibited units in the GPe (**Fig. 6G**), but not in the SNr (**Fig. 6H**). Similarly, local GPe CNO infusion significantly blunted the opto-induced inhibition of GPe units (**Fig. 6I** and **6J** for the first 50 ms), but not of SNr units (**Fig. 6K** and **6L** for the first 50 ms). Since spike frequency in the SNr appears to be affected by CNO early during the inhibition we restricted the analysis to only the first 50 msec of optogenetic stimulation (**Fig. 6J/L**) but did not detect any effect of CNO in the SNr. We confirmed these data using a Z-score based analysis which provided a more stringent criterion for identifying units whose activity was decreased by the optogenetic inhibition (**Supplementary Fig. S6C-H**). Although the Z-score appears slightly decreased in the SNr with CNO, we could not detect any significant effects, either by looking at the full stimulation window or the first 50 ms (**Supplementary Fig. S6D, F, H**). We also identified a low number (0-8) of excited units, but the number was too low for a statistical comparison between the saline and CNO groups (**Supplementary Fig. S7**). Last, while baseline firing in the SNr appears to be lower in CNO injected mice (possibly due to polysynaptic effects), this effect was not significant (**Supplementary Fig. 6A,B**). These in vivo physiology results indicate that local infusion of CNO into the GPe inhibits synaptic release at local dSPN GPe terminals but does not significantly affect action potential propagation in descending dSPN axons going to the SNr. We cannot entirely exclude the existence of possible activity changes in the SNr after GPe CNO infusion. Together with the behavioral data, this supports the notion that dSPN GPe terminals are necessary for normal locomotion and motor control.

6. Npas1-cre/Drd1-cre experiments:

I am hoping the authors can clarify this experimental configuration. First, two BAC transgenic mouse lines were crossed. Did the authors characterize this cross or has it been characterized in a previous publication? While the location of the Drd1-cre transgene is known, it looks like the location of Npas1-cre is unknown, so it is important to confirm that there is not an unwanted phenotype in the double mutant. Also, the Npas1-cre line from MMRRC is listed as also containing tdTomato. Do the Npas1-cre mice also express tdTomato or am I misunderstanding something?

Figure 12: Npas-1-Cre expression pattern within striatum and GPe. (Hernandez, 2015 PMID: 33731445, Fig. 1).

This is a good point. The genomic location of the Drd1-cre BAC transgene and Npas1-cre BAC transgene are, to the best of our knowledge, unknown. The expression pattern of Drd1-cre mice has been well characterized. The expression pattern of Npas-1-Cre mice has been described by our collaborator Savio Chan (PMID 26311767). We choose Npas1-cre mice because they target non-Parvalbumin (PV) positive neurons in the GPe and in contrast to FoxP2-cre mice have no cre expression in the striatum, allowing us to express flex-ChrimsonR in the striatum under the Drd1 promoter (**Revision Letter, Figure 12**). Drd1-cre; Npas1-cre double transgenic mice have no obvious phenotype (PMID 33731445). In the experiment used here we determined if dSPN optogenetic stimulation inhibits Npas1 neurons (we compare a non-stimulated epoch vs. an opto epoch). Our control group were Drd1-cre; Npas1-cre double transgenic mice injected with an mCherry virus; thus all mice had the same genotype. The reviewer is correct that Npas1-cre also expresses TdTomato and we will clarify this in the updated manuscript Methods (Page 22; Line 8) and

Figure 8 Legend. We also explained in the result section why we selected Npas1-cre mice for the experiments (Page 14; Line 41).

Note that TdTomato expression of the Npas1-2A-Tdtomato transgene is too low to be detected without antibody amplification.

Npas1-cre-tdTomato (027718; Jackson; gift from S. Chan)

We here used Drd1-cre mice crossed to Npas1-cre mice. In mice or rats only 55-70% of Npas1 neurons are considered arkypllidal (the other 30-45% project to cortex, reticular thalamus and midbrain instead of striatum)^{20,61-64}; thus FoxP2-cre mice would be more selective for arkypllidal cells than Npas1-cre mice. However, FoxP2 is also expressed in the striatum which would interfere with our dSPN manipulation, while Npas1-cre selectively expresses in the GPe (see Fig. 1 in⁶⁷).

One point raised by the reviewer with this question is if our findings are valid irrespective of the mouse line, or if our findings are only valid for Drd1-cre x Npas1-cre mice. For example, could the strength of the dSPN to Npas1 pathway be altered in Drd1-cre x Npas1-cre mice. In Cui et al. 2021 (PMID 33731445), Drd1-cre x Npas1-cre were used to show, using slice physiology, that dSPNs primarily target Npas1 neurons (partial neuron subpopulation of Arkypllidal cells) rather than PV neurons (partial neuron subpopulation of Prototypical cells). Similar findings were found in mice using *in vivo* physiology in anesthetized Drd1cre mice that did not carry the Npas1-cre transgene (PMID 33248017). Authors showed that dSPNs primarily target Arkypllidal neurons (identified by their physiological properties and FoxP2 expression) rather than Prototypical neurons. Since the data are consistent, this supports the fact that the dSPN-Npas1 circuit is not qualitatively altered in Drd1-cre; Npas1-cre double transgenic mice.

7. Npas1-cre/Drd1-cre experiments:

Second, are the authors certain that GCaMP did not express in dStr axon terminals? The red channel is really saturated in the histology so I can't tell if any GCaMP virus was taken up by axons and (possibly) retrogradely transported.

The reason I ask is that Figure 8D shows an apparent increase right after stim. It's possible this example is unrepresentative, but the increase seems to scale with power (especially in the first 10-30 trials). The authors should carefully look at this data, because a fast-timescale increase would suggest that photometry is both reporting an expected inhibitory response from $Drd1 \rightarrow GPe$ GABA release and the excitatory response in $Drd1$ axon terminals. If the authors performed this control already it would be useful to know what if any GCaMP signal is present in $Drd1$ -cre only animals. If this data is not on hand, a careful re-analysis of their data is hopefully sufficient to resolve this. A good first-step here would be to look at un-z-scored dF/F_0 for the same animal, take the difference between .5/2mW and 0mW and then average?

This is a good point. To avoid the problem of red-channel saturation we present now the images separately per channel. We did not detect GCaMP expression (cyan) in dSPN axons (red) (**Revision Letter, Figure 13**). We added this to manuscript Fig. 8. The tdTomato from the $Npas1$ -cre-TdTomato transgene is too low to be detected without antibody enhancement of the signal. We added a sentence to the figure legend regarding this.

Figure 13: GCaMP expression in $Npas1$ -Cre; $Drd1$ -Cre double transgenic mice (Left) overview at low resolution, (middle) GCaMP expression alone, Chrimson expression alone and overlay, (right) zoom-in at higher resolution. Unstained brain sections.

Note that there is no increase in the fluorescent signal *after* the stimulation starts (time point 0) as the overall quantification shows (manuscript **Fig. 8D**). The increase in fluorescence peaks at time point 0 when the optogenetic stimulation starts. This peak is a consequence of the closed loop approach, waiting for $Npas1$ activity to go up to a threshold before triggering the optogenetic inhibition. We added a sentence to the result section to clarify this point (Page 15; Line 10).

Due to this approach $Npas1$ activity peaks at the time point when the closed-looped stimulation is initiated (**Fig. 8D**).

The visual impression of the heat maps in Fig. 8D is misleading and may be due to the fact that we had originally sorted trials by the level of inhibition (see **Revision Letter, Figure 14A**). If sorted by the order of appearance peaks look comparable between 0, 0.5 and 2 mW; either with 10 s or 3 s optogenetic stimulation. Independent of sorting the important point here is that time point 0 is not followed by an increase in fluorescence as the quantification in Fig. 8D and F show, rather by a decrease. The higher fluorescence around time-point -1.5 to +1.5 sec is due to the closed-loop nature of the experiment, can be seen in all trials (0, 0.5m 2mW) and don't differ across conditions (quantification shown in **Revision Letter, Figure 14B**). This is now more apparent with the unsorted heatmaps (**Revision Letter, Figure 14A**), which we use in the revised paper.

Figure 14: A. Heat maps showing $Npas1$ activity following closed-loop optostimulation of dSPNs for 10 sec (**Fig. 8D**) and 3 sec (**Suppl. Fig. S9**) optogenetic regimens. On the right side we used the same data and sorted them by the order of occurrence. **B.** Average Zscore dFF in the -1.5 to +1.5 s before and after opto stim showing no difference in activity at 0, 0.5 and 2mW trials at the 10 seconds opto experiment ($p = 0.5203$) and 3 sec experiment ($p = 0.8611$). $N=6$ /group.

Figure 15: **A.** Npas1 activity (calcium imaging with photometry) shown using Zscored $\Delta F/F$ (dFF) after closed-loop opto stimulation of dSPNs for 10 sec. Left: heatmap of individual trials, Middle: line curves, Right: Quantification of the amplitude change shows significant decrease in 0.5 mW and 2 mW vs. control 0mW trials. On the right side we used the same data and sorted them by the order of occurrence. **B.** Same data as in A., showing un-z-scored $\Delta F/F$. ** $p < 0.01$, *** $p < 0.001$, N=6/group.

Following the reviewer's suggestion, we also analyzed the un-z-scored $\Delta F/F$ to determine that we did not overlook something. As expected, dSPN optostimulation led to a decrease in Npas1 calcium signal, like for the z-scored data (**Revision Letter, Figure 15**). The zscore-ing does not seem to influence our results. We added a comment about this in the Figure legend (Fig. 8).

Note that we re-analyzed all data without z-scoring and obtain the same results (not shown).

8. ChAT-cre experiments:

In Figure 9 why was ChAT used a comparison and not PV? The expression level in ChAT neurons looks extremely low, so it's hard to tell if the difference in effect is due to cell-type targeting or simply number of cells expressing Chr2.

We used ChAT as a comparison rather than PV because of previous work (PMID 25739505) which showed that dSPNs provide monosynaptic connections to ChAT cells. Recent work (PMID 33731445) also showed that dSPNs provide monosynaptic connections to Npas1 neurons but only limited projections to PV cells and optogenetic stimulation experiments showed that Npas1 neurons are antikinetic while PV cells are prokinetic (PMID 33731450). Thus, PV cells are unlikely to mediate the motor supporting effects of dSPN bridging collaterals. Still, we will discuss that there are limitations when studying Npas1 or PV neurons as an entire population since they can be divided in subpopulations with distinct projection fields. E.g. there are limited PV cells that project to the parafascicular thalamus that receive monosynaptic input from dSPNs and although they did not regulate locomotion (PMID 33723433), it is unclear whether they may contribute to our behavioral effects. We will discuss this now in the discussion on (page 19, line 33):

Furthermore, the contribution of prototypical cells remains to be clarified: indeed, recent work showed that dSPN input to PV neurons (which label most prototypical cells⁷⁶) is weak¹⁶ and that PV neurons inhibit locomotion⁶⁵. Thus if dSPN bridging collaterals would act via inhibiting PV neurons, PV neurons should decrease and not enhance locomotion. On the other hand, a recent paper showed that bridging collaterals dSPNs target a small population of Lhx6-negative parafascicular thalamus-projecting PV cells, but not STN/SNr-projecting PV cells, which regulate reversal learning, but not locomotion. Moreover, a small population of Lhx6+ cells that are negative for PV and Npas1^{62,67} have not been tested as to whether they receive dSPN input and their role in motor control is unknown. Finally, we observed here that ChAT neurons, which are targeted by dSPNs⁶⁸, do not recapitulate the motor effects of bridging collaterals. It is possible, however, that ChAT neurons mediate other behavioral effects of dSPN collaterals not addressed here.

We think that the low expression is because the density of ChAT cells at the GPe border is very small, as shown in (PMID 25739505). We had to restrict expression to the GPe and avoid the neighboring nucleus basalis where there are also ChAT cells that do not receive dSPN input (PMID 25739505). It is possible that low expression could explain the lack of motor effect. Nevertheless, even a small population of GPe ChAT neurons may modulate behavior as they have broad projections to the cortex evoking ACh and GABA mediated currents in the cortex (PMID 25739505: Extended Fig 7).

9. Minor points:

Photometry experiments are referred to as “imaging”. Since photometry is not image forming so I would replace “calcium imaging” with “calcium photometry”.

This is a good point. We are happy to edit “calcium imaging” to “calcium recordings with fiber photometry” or “calcium photometry”; which we have now done in the revised paper.

10. Minor points:

Virus titers are not reported. This is important to know to enhance reproducibility and is an important detail since the interpretation of the results depends on ruling out any “leaky” expression of DIO constructs.

Titers were in a separate Excel sheet document called **Supplementary Table 2** but this may have been difficult to find. We moved it to the Supplementary Material word document and include it as an embedded Table so that they are easier to find for the reader. We agree that it is important that they are readily available. We also added a comment in the Methods to say where titers can be found.

11. Minor points:

Checking to see if this is a typo, in Fiber photometry during behavior in methods the authors mention demodulating offline then applying a 3 Hz low pass filter and sampling the 3 Hz low-passed signal at 1017.3 Hz. Do I have that right that the signals were this over-sampled?

Indeed, this is a high sampling rate and it was not a typo. The signals were over-sampled at data collection at 1017.3 Hz, demodulated, then the processor applied a 3Hz low-pass filter. We use a processor from Tucker-Davis Technologies which requires this high sampling rate for optimal signal frequency-based modulation (at 300-500Hz) and demodulation. Since this high sampling rate is not required for data analysis, we later downsample 10 times (offline) to 100 Hz.

12. Minor points:

Related to the previous point, another smoothing filter is applied to the photometry data, either 1 H low-pass z or a 0.5s moving average. This sounds like the photometry signals are really smoothed. I’m surprised that the 3 Hz low-pass was not enough to remove high frequency noise. Can authors comment on this?

The 3 Hz low-pass filter successfully removed most of the high frequency noise, however, residual low amplitude fast fluctuations were still present in the datasets “riding” on top of the behaviorally aligned (“jumps on the rotarod”) slower fluctuations on the ~1 Hz or lower frequency range. As we attempted to better link the neural data with behavior, we performed a peak detection approach in both photometry and behavioral datasets (see **Supplementary Fig. S4D,E**). Since the residual low amplitude fast fluctuations made the peak detection in the photometry data more arduous and required establishing more arbitrary parameters for including or excluding individual peaks on the analysis, we decided to apply a 0.5s moving average smoothing. This approach successfully removed the residual low amplitude high frequency fluctuations while preserving the main and behaviorally-relevant signal (See response to point 4; **Revision Letter, Figure 8**).

13. Minor points:

For all filters used in data pre-processing please specify the filter that was used and how it was applied to the data (e.g. if it was applied forward and backwards to remove phase distortions).

We used a 3-Hz low-pass filter built into the data collection software (Synapse) which is a one-way 2nd order Butterworth filter. Additionally for peak detection analyses of (**Supplementary Fig S4D,E**), we applied a 0.5s moving average smoothing which did not distort the data (**Revision Letter, Figure 8**). We added this to the Methods section (page 23, line 20 and page 26, line 13).

Signals were sinusoidally modulated, using Synapse software and RZ5P Multi I/O Processors (Tucker-Davis Technologies), at 210 and 330 Hz (405 and 465 nm, respectively) via a lock-in amplification detector, then demodulated and low-passed filtered at 3 Hz on-line (one-way 2nd order Butterworth filter).

For peak frequency and amplitude and interpeak interval calculation presented in **Supplementary Figure S4D-E**, dFF was low-pass filtered using a 0.5s moving average smoothing to avoid overcounting peaks not relevant to the behavior of interest (jumps). We verified the filter did not alter the data (data not shown).

REVIEWERS' COMMENTS

Reviewer #1 (Remarks to the Author):

The authors have addressed all my comments, as well as those of the other reviewer. Consequently, the paper has been significantly improved, and I extend my congratulations to the authors for conducting this very nice work.

Reviewer #2 (Remarks to the Author):

Through extensive experiments and analysis the authors have gone to great lengths to address all of my comments from the last round. At this point all of my major concerns have been addressed. I applaud the authors on their findings and think this will be a welcome contribution to Nature Communications.